# SSR-Merge: Subspace Signal Routing
# for Training-Free LoRA Merging in Diffusion Models

**Zhengxuan Wei** [1 2 3]  **Yi Dong** [2]  **Zonghui Li** [2 4]  **Xianhui Lin** [2]  **Xing Liu** [2]  **Hong Gu** [2]  **Shaofeng Zhang** [5]
**Wenbin Li** [1]  **Qi Fan** [1]

## Abstract

Low-Rank Adaptation (LoRA) merging can efficiently combine diverse generative capabilities from multiple trained LoRAs for a diffusion model. However, existing LoRA merging techniques often suffer from severe parameter interference, causing destructive collisions in the shared parameter space. To address this, we propose Subspace Signal Routing (SSR), which resolves interference by routing internal signals instead of performing parameter-space merge. Specifically, SSR first constructs a unified subspace by concatenating candidate LoRAs along the rank dimension. Next, SSR employs an inverse correlation matrix to decorrelate mixed signals within this space. Finally, a directional guide matrix steers these purified signals into their respective task-specific subspaces. We provide a rigorous theoretical analysis proving that SSR aligns with the Ordinary Least Squares (OLS) solution, thereby ensuring mathematical optimality. We utilize the additivity of sufficient statistics to design a streaming algorithm. This enables on-the-fly updates that significantly reduce memory overhead and computation time. Extensive experiments validate that SSR significantly outperforms state-of-the-art methods while maintaining comparable efficiency. The source code will be made publicly available.

## 1. Introduction

Diffusion models demonstrate remarkable capabilities in synthesizing high-fidelity and diverse visual content (Ho

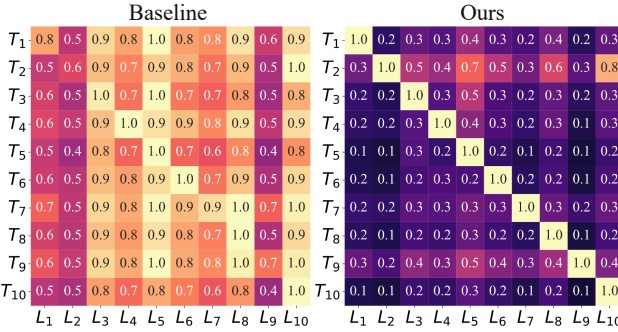

*Figure 1.* **Visualization of Task-LoRA Activation Alignment.** We display the max-normed activation intensity between task instructions ($T$) and LoRA modules ($L$). **Left:** The static baseline exhibits severe crosstalk, where instructions spuriously activate unrelated modules, indicating high interference. **Right:** Our SSR-Merge achieves precise signal routing, showing a clean diagonal structure where each task exclusively activates its target LoRA. Please refer to Appendix A for the detailed setup.

et al., 2020; Song et al., 2021; Rombach et al., 2022; Ramesh et al., 2022; Saharia et al., 2022). However, full-parameter fine-tuning for downstream adaptation is computationally expensive (Wiggins & Tejani, 2022). To address this, Parameter-Efficient Fine-Tuning (PEFT) techniques have emerged as a standard paradigm, enabling effective adaptation while updating only a fraction of the model parameters (Hu et al., 2022; Houlsby et al., 2019; Li & Liang, 2021; Lester et al., 2021; Zhang et al., 2023).

Among various PEFT approaches, Low-Rank Adaptation (LoRA) (Hu et al., 2022) is the most prominent, balancing efficiency and quality by injecting trainable low-rank matrices into frozen layers. This efficiency has fostered a vast ecosystem of specialized LoRAs across model hubs (Wolf et al., 2020), spanning diverse styles, characters, and instructions. This naturally leads to a need for composition, where users seek to combine distinct capabilities within a single model by merging multiple LoRAs.

However, integrating multiple task-specific LoRAs remains a significant challenge. Basic static strategies, such as linear averaging (Wortsman et al., 2022) and Task Arithmetic (Ilharco et al., 2023), directly superimpose parameters, inher-

[1]School of Intelligence Science and Technology, Nanjing University, China [2]vivo BlueImage Lab, vivo Mobile Communication Co., Ltd., China [3]ShanghaiTech University [4]Southeast University [5]University of Science and Technology of China. Correspondence to: Qi Fan <fanqi@nju.edu.cn>, Yi Dong <ydong@outlook.com>.

*Proceedings of the 43[rd] International Conference on Machine Learning*, Seoul, South Korea. PMLR 306, 2026. Copyright 2026 by the author(s).

ently causing parameter interference. To address this, heuristic variants like TIES (Yadav et al., 2023) and DARE (Yu et al., 2024) attempt to prune conflicting weights. However, they fail to prevent destructive collisions as the number of tasks scales. As visualized in Figure 1 (Left), the DARE baseline exhibits severe "crosstalk", where instructions spuriously activate unrelated modules with high intensity.

Alternatively, dynamic approaches (Mao et al., 2022; Wang et al., 2024a; Zhao et al., 2024; Wu et al., 2024) focus on assigning adaptive weights to combined modules. However, methods learning scalar coefficients (Wang et al., 2024a; Zhao et al., 2024) struggle to resolve conflicts within the shared parameter space. Conversely, approaches utilizing non-linear gating (Mao et al., 2022; Wu et al., 2024) break the re-parameterization property, preventing weight merging and incurring inference latency.

To address this, we propose Subspace Signal Routing (SSR). Instead of performing direct arithmetic operations in the parameter space, SSR avoids conflicts by explicitly routing the internal signals within the subspace of the LoRA.

As illustrated in Figure 2, SSR first constructs a unified subspace by concatenating candidate LoRAs along the rank dimension. Within this subspace, we insert a training-free Router ($R$) derived from second-order statistics. Specifically, the router employs an inverse correlation matrix ($\mathbf{G}^{-1}$) to act as a whitening filter that decorrelates mixed intermediate signals, followed by a directional guide ($\mathbf{Q}$) that precisely steers these purified signals into their respective task-specific subspaces. This mechanism effectively eliminates inter-task interference, as evidenced by the clean diagonal activation pattern in Figure 1 (Right), demonstrating that SSR successfully disentangles conflicting signals and ensures precise task activation.

Crucially, this routing mechanism is backed by a rigorous proof of optimality. We demonstrate that the router is mathematically equivalent to the projection of the Ordinary Least Squares (OLS) estimator. This formulation ensures that SSR provides the unique analytical solution that strictly minimizes the feature reconstruction error, rather than relying on heuristic parameter tuning.

To ensure computational feasibility, we design a streaming algorithm grounded in the additivity of sufficient statistics. By accumulating covariance and cross-correlation updates on-the-fly, we eliminate the need to cache raw features, effectively reducing memory complexity. For deployment, we employ structural re-parameterization to absorb the router into the up-projection weights. This yields a merged module structurally identical to standard LoRA, guaranteeing no additional inference overhead and seamless compatibility with existing ecosystems (von Platen et al., 2022).

Our contributions are summarized as follows:

- We introduce SSR, reframing model merging as signal routing rather than parameter arithmetic. It utilizes a unified subspace to statistically decorrelate and steer signals, thereby eliminating interference.

- We theoretically prove that our router is analytically equivalent to the projection of the Ordinary Least Squares (OLS) estimator, thereby strictly minimizing the feature reconstruction error.

- We develop a streaming algorithm and structural re-parameterization to ensure efficient deployment. Experiments demonstrate that SSR achieves superior capability preservation across diverse tasks.

## 2. Related Work

**Model Merging.** Model merging has evolved from simple weight averaging (Wortsman et al., 2022) to Task Arithmetic's (Ilharco et al., 2023) composition of task vectors. However, direct arithmetic often suffers from parameter interference. To mitigate this, TIES-Merging (Yadav et al., 2023) resolves conflicts via trimming and sign election, while DARE (Yu et al., 2024) establishes a standard baseline by employing random sparsification and rescaling to approximate original expectations. Beyond these mainstream paradigms, recent research has explored other directions, including masking and outlier-aware strategies (Davari & Belilovsky, 2024; Wang et al., 2024b), spectral and representation alignment schemes that utilize geometric decompositions or feature matching (Stoica et al., 2025; Gargiulo et al., 2025; Marczak et al., 2025; Panariello et al., 2025; Yang et al., 2024a; Ainsworth et al., 2023; Stoica et al., 2024), and adaptive weighting frameworks that estimate mixing coefficients through iterative search or analytical computation (Jin et al., 2023; Matena & Raffel, 2022; Huang et al., 2023; Yang et al., 2024b; Chen et al., 2025b; Huang et al., 2024; Lu et al., 2024).

While generic strategies often neglect the underlying parameter structure, our SSR explicitly leverages the intrinsic geometry of low-rank subspaces. We provide a framework backed by theoretical analysis, which resolves interference and preserves adapter integrity more effectively than naive weight arithmetic.

**Applications of LoRA Merging.** LoRA composition is widely adopted across machine learning domains. In LLMs and distributed systems, it facilitates multi-task generalization and federated learning (Wu et al., 2024; Huang et al., 2023; Chen et al., 2023). In diffusion models, research primarily focuses on specific scenarios, such as subject-style fusion (Shah et al., 2024; Frenkel et al., 2024; Shenaj et al., 2025; Ouyang et al., 2025; Roy et al., 2025) or multi-concept composition (Gu et al., 2023; Yang et al., 2025).

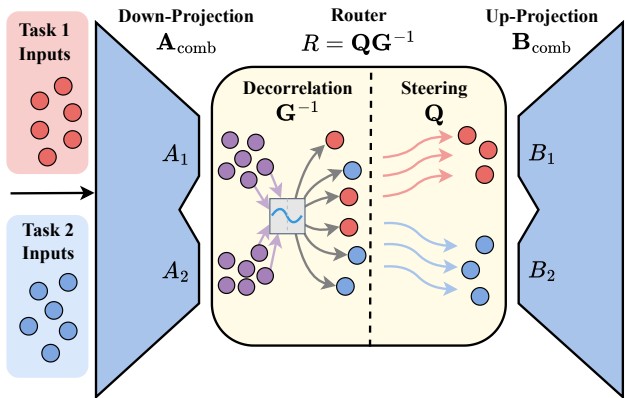

*Figure 2.* **Overview of Subspace Signal Routing (SSR).** The framework expands individual LoRA bottlenecks into a unified subspace via $\mathbf{A}_{\mathrm{comb}}$. Within this space, the **Router** $R$ resolves parameter interference through a two-stage mechanism: (1) **Decorrelation** ($\mathbf{G}^{-1}$), which acts as a decorrelation operator to disentangle mixed intermediate features; and (2) **Steering** ($\mathbf{Q}$), which precisely guides the purified signals towards their target task-specific up-projection bases ($B_1, B_2$). Ideally, this linear structure allows multiple tasks to coexist without conflict.

Given the rapid proliferation of diverse adapters, there is a growing demand for a unified merging framework. To address this, our SSR offers a general-purpose, training-free solution that resolves interference without relying on task priors or complex architectural designs.

## 3. Methodology

We introduce Subspace Signal Routing (SSR) to resolve multi-task interference. The section is organized into problem formulation (Sec. 3.1), router construction (Sec. 3.2), theoretical analysis of optimality (Sec. 3.3), and efficient implementation (Sec. 3.4).

### 3.1. Preliminary

Low-Rank Adaptation (LoRA) (Hu et al., 2022) approximates weight updates $\Delta W$ for a pre-trained matrix $W_0 \in \mathbb{R}^{d \times d}$ via low-rank decomposition. Let $A \in \mathbb{R}^{r \times d}$ and $B \in \mathbb{R}^{d \times r}$ denote the down- and up-projection matrices, respectively, where $r \ll d$. The adapted forward pass for an input activation $x \in \mathbb{R}^d$ is formulated as:

$$h = W_0 x + \Delta W x = W_0 x + B A x. \tag{1}$$

We consider the problem of merging $K$ distinct LoRA modules $\{A_k, B_k\}_{k=1}^K$ trained on different tasks. Our objective is to synthesize a single unified update $\Delta W_{\mathrm{merged}}$ that preserves the capabilities of all $K$ tasks while minimizing mutual interference.

### 3.2. Subspace Signal Routing

To resolve parameter interference among the $K$ tasks, we propose Subspace Signal Routing (SSR). This method builds a unified space where conflicting features are separated via a routing matrix $R$.

**Unified Projection Space.** We first combine the individual subspaces into a unified coordinate system. We construct a unified down-projection $\mathbf{A}_{\mathrm{comb}} \in \mathbb{R}^{Kr \times d}$ by vertically stacking the task-specific down-projections $A_k$, and a unified up-projection $\mathbf{B}_{\mathrm{comb}} \in \mathbb{R}^{d \times Kr}$ by horizontally concatenating the up-projections $B_k$:

$$\mathbf{A}_{\mathrm{comb}} = \begin{bmatrix} A_1 \\ \vdots \\ A_K \end{bmatrix}, \quad \mathbf{B}_{\mathrm{comb}} = \begin{bmatrix} B_1 & \dots & B_K \end{bmatrix}. \tag{2}$$

This formulation expands the rank from $r$ to $Kr$. Our goal is to design a router matrix $R \in \mathbb{R}^{Kr \times Kr}$ inserted between $\mathbf{A}_{\mathrm{comb}}$ and $\mathbf{B}_{\mathrm{comb}}$ to control the signal flow.

**Routing via Statistics.** We design the router using second-order statistics derived from calibration data. Let $X_k \in \mathbb{R}^{d \times N}$ be the input features for the $k$-th task, and $Z_k = \mathbf{A}_{\mathrm{comb}} X_k \in \mathbb{R}^{Kr \times N}$ be its projection in the unified space.

First, the **Correlation matrix** $\mathbf{G} \in \mathbb{R}^{Kr \times Kr}$ measures the total correlation structure in the combined space, defined as the sum of outer products of the projected features:

$$\mathbf{G} := \sum_{k=1}^K Z_k Z_k^\top. \tag{3}$$

Second, the **Directional Guide** $\mathbf{Q} \in \mathbb{R}^{Kr \times Kr}$ captures the cross-covariance between the unified projections and the task-specific targets. For the $k$-th task, with the ideal local activation $H_k = A_k X_k$, we define the task-specific routing block $\mathbf{Q}_k = H_k Z_k^\top$. The global guide $\mathbf{Q}$ is constructed by stacking these blocks:

$$\mathbf{Q} := \begin{bmatrix} \mathbf{Q}_1 \\ \vdots \\ \mathbf{Q}_K \end{bmatrix} = \begin{bmatrix} (A_1 X_1) Z_1^\top \\ \vdots \\ (A_K X_K) Z_K^\top \end{bmatrix}. \tag{4}$$

**Router Formulation.** Based on these statistics, we construct the Subspace Router $R$ by combining the directional guide with the inverse energy map:

$$R := \mathbf{Q} \mathbf{G}^{-1}. \tag{5}$$

In this design, $\mathbf{G}^{-1}$ acts as a decorrelation operator to remove the correlation in the shared signal space, while $\mathbf{Q}$ guides these purified signals towards the up-projection bases $B_k$ of their respective tasks.

## 3.3. Theoretical Analysis of Optimality

In this section, we reveal that the proposed router $R$ is not merely a heuristic design, but the exact analytical solution that minimizes the reconstruction error across all tasks.

**Geometric Decomposition.** To understand the mathematical behavior of the router, we analyze the structure of the Directional Guide $\mathbf{Q}$. By the property of the Moore-Penrose pseudoinverse, we have:

$$\mathbf{B}_{\text{comb}}^{\dagger} B_k = \mathbf{E}_k, \tag{6}$$

where $\mathbf{E}_k$ is the canonical block selection matrix. Applying this geometric property to the definition of $\mathbf{Q}$, we can substitute the selection matrix with the projection operator:

$$\mathbf{Q} = \sum_{k=1}^{K} \mathbf{E}_k (A_k X_k) Z_k^{\top} \\ = \sum_{k=1}^{K} (\mathbf{B}_{\text{comb}}^{\dagger} B_k)(A_k X_k) Z_k^{\top}. \tag{7}$$

Since the inverse projector $\mathbf{B}_{\text{comb}}^{\dagger}$ is invariant to the task index $k$, it factors out of the summation. Recognizing that $B_k A_k X_k$ corresponds to the target signal $Y_k$, we arrive at the unified form:

$$\mathbf{Q} = \mathbf{B}_{\text{comb}}^{\dagger} \sum_{k=1}^{K} \underbrace{(B_k A_k X_k)}_{Y_k} Z_k^{\top} = \mathbf{B}_{\text{comb}}^{\dagger} \left( \sum_{k=1}^{K} Y_k Z_k^{\top} \right). \tag{8}$$

**Equivalence to Least Squares.** Substituting this factorized form back into the router definition $R = \mathbf{Q}\mathbf{G}^{-1}$, the expression simplifies significantly:

$$R = \mathbf{B}_{\text{comb}}^{\dagger} \underbrace{\left( \sum_{k=1}^{K} Y_k Z_k^{\top} \right) \left( \sum_{k=1}^{K} Z_k Z_k^{\top} \right)^{-1}}_{\hat{\beta}_{\text{OLS}}}. \tag{9}$$

Here, we identify the term $\hat{\beta}_{\text{OLS}}$ as the standard Ordinary Least Squares estimator. In statistical signal processing, $\hat{\beta}_{\text{OLS}}$ is rigorously defined as the unique matrix that minimizes the regression residual $\|\beta Z - Y\|_F^2$.

**Optimality Conclusion.** The derivation above leads directly to our main theoretical guarantee.

**Theorem 3.1** (Reconstruction Optimality). *The Subspace Router $R$ minimizes the reconstruction objective $\mathcal{L}(R) = \sum_{k=1}^{K} \|\mathbf{B}_{comb} R Z_k - Y_k\|_F^2$.*

*Proof.* As derived, our router implements the projection of the optimal estimator:

$$R = \mathbf{B}_{\text{comb}}^{\dagger} \hat{\beta}_{\text{OLS}}. \tag{10}$$

Consequently, the merged model output is given by $\hat{Y} = \mathbf{B}_{\text{comb}} R Z$. Substituting the router form:

$$\hat{Y} = (\mathbf{B}_{\text{comb}} \mathbf{B}_{\text{comb}}^{\dagger}) \hat{\beta}_{\text{OLS}} Z. \tag{11}$$

Since the targets $Y_k$ (and thus $\hat{\beta}_{\text{OLS}}$) lie within the range of the up-projection $\mathbf{B}_{\text{comb}}$, the projection operator $\mathbf{B}_{\text{comb}} \mathbf{B}_{\text{comb}}^{\dagger}$ acts as an identity mapping, yielding $\hat{Y} = \hat{\beta}_{\text{OLS}} Z$. Since $\hat{\beta}_{\text{OLS}}$ is defined as the minimizer of the Frobenius norm discrepancy, $R$ inherently achieves the minimal possible reconstruction error. □

## 3.4. Efficient Implementation

**Streaming Calculation.** A naive offline construction necessitates caching activation maps for all $K$ tasks across the entire model, resulting in a prohibitive memory complexity of $\mathcal{O}(K \cdot N_{\text{layer}} \cdot T \cdot D_{\text{feat}})$, where $N_{\text{layer}}$ denotes the total number of LoRA layers, $T$ the calibrated timesteps, and $D_{\text{feat}}$ the feature volume per layer. To address this, we employ an exact streaming strategy grounded in the additive property of sufficient statistics. For an incoming feature batch $x_t$ belonging to task $k$, we update the statistics on-the-fly:

$$\mathbf{G} \leftarrow \mathbf{G} + (\mathbf{A}_{\text{comb}} x_t)(\mathbf{A}_{\text{comb}} x_t)^{\top} \tag{12}$$
$$\mathbf{Q} \leftarrow \mathbf{Q} + \mathbf{E}_k (A_k x_t)(\mathbf{A}_{\text{comb}} x_t)^{\top} \tag{13}$$

where $\mathbf{E}_k$ places the vector into the $k$-th block of the stacked space. The raw feature batch $x_t$ is discarded after the update. This guarantees numerical equivalence to the solution while reducing the space complexity to a constant $\mathcal{O}((Kr)^2)$.

**Structural Re-parameterization.** For deployment, we exploit linearity to absorb the router $R$ into the up-projection:

$$\tilde{\mathbf{B}}_{\text{comb}} = \mathbf{B}_{\text{comb}} R. \tag{14}$$

The resulting $(\mathbf{A}_{\text{comb}}, \tilde{\mathbf{B}}_{\text{comb}})$ remains structurally identical to standard LoRA, ensuring seamless compatibility with existing frameworks. Moreover, this form allows the update to be fully merged into the backbone weights, thus achieving strict zero inference latency.

**Overall Pipeline.** The complete pipeline integrating these strategies is summarized in Algorithm 1.

## 3.5. Data Construction and Analysis

**One-Shot Calibration.** We obtain calibration data without relying on external datasets. Instead, we assign a single representative input for each task. For instance, given a LoRA trained on a specific dog concept, we simply use a text prompt like ``A [V] dog'', crucially without requiring any ground-truth images. The input features to the LoRA modules are directly extracted as $X_k$. Notably, while generative models typically involve multi-step inference, we perform forward propagation for only a single timestep to

**Algorithm 1** Streaming Subspace Signal Routing (SSR)

---

**Input:** Task LoRAs $\{(A_k, B_k)\}_{k=1}^K$, Calibration Data $\mathcal{D}$
**Output:** Merged Model $(\mathbf{A}_{\text{comb}}, \tilde{\mathbf{B}}_{\text{comb}})$
  1: *// 1. Initialization*
  2: Construct global projections: $\mathbf{A}_{\text{comb}} \leftarrow [A_1; \ldots; A_K]$, $\mathbf{B}_{\text{comb}} \leftarrow [B_1, \ldots, B_K]$
  3: Initialize statistics: $\mathbf{G} \leftarrow \mathbf{0}, \mathbf{Q} \leftarrow \mathbf{0}$
  4: *// 2. Streaming Accumulation*
  5: **for** task $k = 1$ **to** $K$ **do**
  6:    Load LoRA module $(A_k, B_k)$ into base model
  7:    **for** input batch $x_t$ from $\mathcal{D}$ **do**
  8:       $z_t \leftarrow \mathbf{A}_{\text{comb}} x_t$
  9:       $\mathbf{G} \leftarrow \mathbf{G} + z_t z_t^\top$
 10:       $\mathbf{Q} \leftarrow \mathbf{Q} + \mathbf{E}_k(A_k x_t) z_t^\top$
 11:    **end for**
 12:    Unload LoRA module $(A_k, B_k)$
 13: **end for**
 14: *// 3. Router Construction & Re-parameterization*
 15: $R \leftarrow \mathbf{Q}\mathbf{G}^{-1}$
 16: $\tilde{\mathbf{B}}_{\text{comb}} \leftarrow \mathbf{B}_{\text{comb}} R$
 17: **return** $(\mathbf{A}_{\text{comb}}, \tilde{\mathbf{B}}_{\text{comb}})$

---

construct $R$ (see Appendix G for empirical validation and theoretical justification on why a single timestep suffices).

**Statistical Sufficiency.** Despite the one-shot setting, the aggregated feature sequence provides an effective sample size $N$ (total tokens) on the order of $10^3$. This significantly exceeds the subspace dimension ($N \gg Kr$), ensuring that the correlation matrix $\mathbf{G}$ is well-conditioned and invertible. Moreover, as proven in Appendix H, the estimation error is bounded by $\mathcal{O}(\sqrt{Kr/N})$, guaranteeing a tight approximation of the optimal router.

## 4. Experiment

In this section, we conduct a comprehensive evaluation to validate the effectiveness of Subspace Signal Routing (SSR). Our experiments are designed to answer the following three core research questions:

1. **RQ1:** Can the merged LoRA effectively preserve the performance of the individual single LoRA?

2. **RQ2:** Can the merged model execute instructions that require multiple LoRA capabilities simultaneously?

3. **RQ3:** Is the proposed method effective across different types of diffusion tasks, such as image editing?

### 4.1. RQ1: Single-Task Capability Preservation

**Objective and Protocol.** To evaluate single-task preservation, we adopt a variable-scale merging protocol. We draw

from the pool of 10 LoRAs to construct merged models with varying task counts $K \in \{1, 3, 5, 7, 9\}$. Specifically, for a specific target task, we merge its LoRA with $K - 1$ randomly selected "distractor" LoRAs and evaluate the merged model's ability to generate the target subject. The case $K = 1$ represents the Single LoRA baseline (i.e., no merging), serving as the performance upper bound (Oracle). As $K$ increases from 3 to 9, the merged model faces intensifying parameter interference, rigorously testing the robustness of the merging method.

**Implementation.** We conduct experiments using FLUX.1-dev (Labs, 2024) as the foundation model. All experiments are performed on a single NVIDIA A100 GPU. We curate a benchmark of 10 distinct objects from the DreamBooth dataset (Ruiz et al., 2023), selected to maximize structural and textural diversity (see Appendix B.1 for the full list). For each subject, we train a dedicated LoRA with rank $r = 32$, learning rate $1e{-}4$, and 500 training steps. To further verify architectural generality, we additionally evaluate SSR on Qwen-Image (Wu et al., 2025) in Appendix F.

**Baselines.** We benchmark SSR against standard training-free merging paradigms, including Linear Average (Wortsman et al., 2022) and Task Arithmetic (Ilharco et al., 2023). Notably, we demonstrate in Appendix B.2 that Task Arithmetic is mathematically equivalent to a special case of our framework where the routing matrix is the identity ($R = I$), thereby serving as a naive ensemble baseline without signal regulation. We also compare with state-of-the-art interference-resolution methods: TIES-Merging (Yadav et al., 2023), DARE (Yu et al., 2024), the PEFT-specific RobustMerge baseline (Zeng et al., 2025), and IterIS (Chen et al., 2025a). Details for all baselines are provided in Appendix B.2. Additionally, we evaluated RegMean (Jin et al., 2023), but found it to suffer from severe numerical instability in this setting; a detailed analysis is provided in Appendix E.

**Rank Budget Fairness.** SSR does not use a larger subspace capacity than the arithmetic and sparsification baselines. Directly summing $K$ LoRA updates can be exactly written as a rank-concatenated LoRA:

$$\sum_{k=1}^K B_k A_k = \begin{bmatrix} B_1 & \ldots & B_K \end{bmatrix} \begin{bmatrix} A_1 \\ \vdots \\ A_K \end{bmatrix}. \quad (15)$$

This corresponds to our framework with the identity router $R = \mathbf{I}_{Kr}$. Thus, Task Arithmetic, TIES, DARE, and SSR operate on the same collection of $K$ low-rank updates; SSR only replaces identity routing with a statistics-derived router.

**Metrics.** To quantitatively assess knowledge preservation, we measure the fidelity of the generated images against the original ground truth training images of the specific subject. Since each subject corresponds to multiple reference images,

*Table 1.* Quantitative evaluation of single-task capability preservation on FLUX.1-dev under multi-task interference. We report the average DINOv2 and CLIP scores across varying numbers of merged tasks ($K$). The Upper Bound (shown in gray) represents the performance of a standalone LoRA without merging.

| Method | K=3 | | K=5 | | K=7 | | K=9 | |
|---|---|---|---|---|---|---|---|---|
| | DINO | CLIP | DINO | CLIP | DINO | CLIP | DINO | CLIP |
| Average | 0.4374 | 0.7400 | 0.3808 | 0.6944 | 0.3973 | 0.7095 | 0.3739 | 0.7103 |
| Task Arithmetic | 0.5814 | 0.7395 | 0.4935 | 0.7188 | 0.5165 | 0.6546 | 0.5356 | 0.6831 |
| TIES | 0.6264 | 0.7117 | 0.5058 | 0.6953 | 0.5095 | 0.6986 | 0.4723 | 0.6839 |
| DARE | 0.7171 | 0.7574 | 0.6584 | 0.7002 | 0.6087 | 0.7464 | 0.5837 | 0.7376 |
| RobustMerge | 0.7220 | 0.7710 | 0.6670 | 0.7550 | 0.5910 | 0.7480 | 0.5480 | 0.7330 |
| IterIS | 0.7030 | 0.7800 | 0.6720 | 0.7660 | 0.6420 | 0.7580 | 0.6240 | 0.7520 |
| **SSR (Ours)** | **0.7342** | **0.8144** | **0.7059** | **0.7951** | **0.6868** | **0.7798** | **0.6713** | **0.7850** |
| *Recovery Rate* | 98.6% | 96.4% | 94.8% | 94.1% | 92.3% | 92.3% | 90.2% | 92.9% |
| *Upper Bound* | *0.7443* | *0.8452* | *0.7443* | *0.8452* | *0.7443* | *0.8452* | *0.7443* | *0.8452* |

we compute the similarity against all references and report the average. We employ two metrics: (1) CLIP-Score (Radford et al., 2021) to evaluate high-level semantic alignment, and (2) DINOv2 (Oquab et al., 2023) Similarity to assess fine-grained visual identity and structural details. All methods utilize the same initial noise seeds for fair comparison.

**Quantitative Results.** Table 1 presents the evaluation results on FLUX.1-dev under varying interference levels. First, regarding absolute performance in high-interference settings ($K = 9$), SSR consistently outperforms the strongest baselines, exceeding IterIS, the strongest baseline in this setting, by 0.0473 DINO and 0.0330 CLIP. Second, concerning robustness to scaling, baselines degrade significantly as task counts increase, whereas SSR remains stable across all merging scales. Finally, in terms of fidelity retention, SSR consistently recovers over 90.2% of the single-task oracle performance on FLUX.1.

**Qualitative Results.** Figure 3 illustrates the visual fidelity of different merging methods on FLUX.1-dev. Linear Average and Task Arithmetic exhibit semantic drift as $K$ increases. In the Dog column, the specific traits of the Corgi morph into a generic husky-like appearance. TIES and DARE exhibit attribute mismatch and concept leakage. For the RC Car task, these methods fail to preserve the original red-and-yellow colors, often rendering the object in blue. In the Red Cartoon scenario, these baselines introduce unrelated elements such as Santa Claus or realistic sketches into the flat character domain. In contrast, SSR maintains the subject identity, correct attribute binding, and stylistic integrity across all tested merging scales.

**Efficiency Analysis.** Table 2 reports the wall-clock time required for the merging phase. Benefiting from the one-shot calibration strategy, SSR requires only a single inference step to accumulate sufficient statistics. Consequently, under the most demanding setting ($K = 9$), SSR completes

*Table 2.* Comparison of the total wall-clock time (in seconds) to merge $K$ LoRAs across all transformer blocks of FLUX.1-dev. Darker shades indicate slower processing speeds. SSR achieves high efficiency comparable to lightweight arithmetic methods.

| Method | $K = 3$ | $K = 5$ | $K = 7$ | $K = 9$ |
|---|---|---|---|---|
| TIES | 42.78 | 50.98 | 69.87 | 88.93 |
| DARE | 6.96 | 11.75 | 16.11 | 20.95 |
| SSR | 9.50 | 16.31 | 25.46 | 34.26 |

the process in 34.26 seconds. This performance is approximately $2.6\times$ faster than the optimization-based method TIES (88.93 seconds). Compared to the arithmetic baseline DARE (20.95 seconds), SSR incurs an overhead of 13.31 seconds, maintaining a comparable magnitude of efficiency while providing superior interference resolution.

### 4.2. RQ2: Simultaneous Multi-Task Execution

**Objective and Protocol.** We design a Multi-Concept Composition Protocol to evaluate whether the merged model can execute instructions requiring multiple LoRA capabilities simultaneously. This protocol focuses on generating multiple distinct subjects in a single image by merging $K$ randomly sampled LoRAs, with $K$ uniformly distributed in $\{2, 3, 4\}$. The model is then prompted with composite instructions containing trigger words for all $K$ subjects. This configuration assesses the ability of the merged model to represent all requested subjects accurately while preserving their specific visual identities.

**Implementation.** We utilize FLUX.1-dev (Labs, 2024) and the 10-subject LoRA pool described in Sec. 4.1. To assess compositional capabilities, we programmatically generate 100 distinct prompts with task counts K uniformly distributed across $\{2, 3, 4\}$, ensuring a systematic evaluation of multi-task performance. We evaluate SSR against

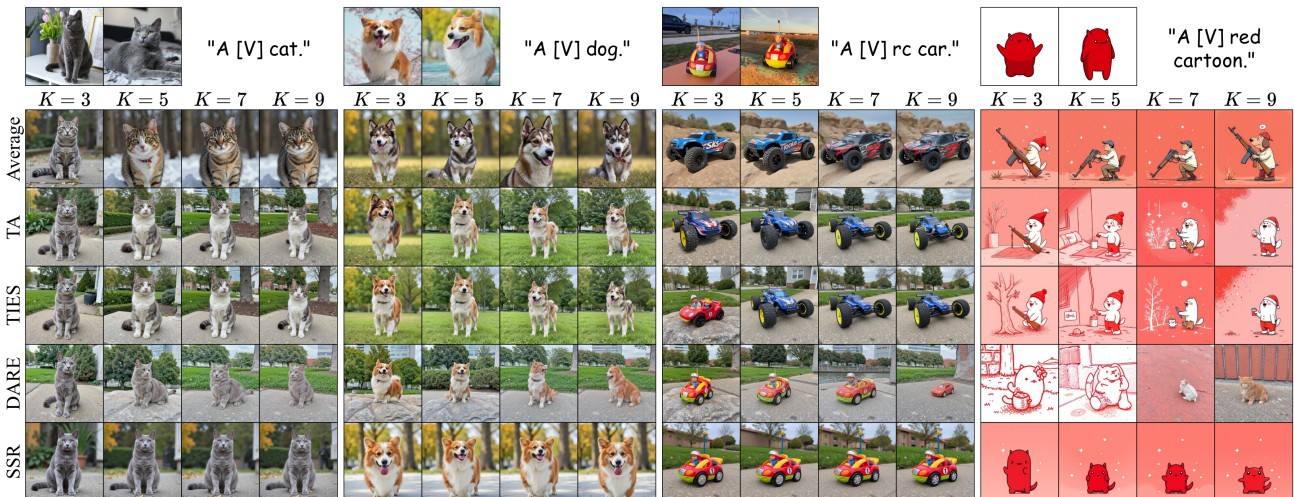

*Figure 3.* Visual comparison of single-task generation results under increasing merging scales with K ranging from 3 to 9. The top row displays ground truth reference images and text prompts, where [V] represents the learned unique identifier token for each specific subject. Subsequent rows present the generated outputs from Linear Average, Task Arithmetic, TIES-Merging, DARE, and SSR. For each column, the unified model containing K task-specific LoRAs is prompted to generate the target subject shown in the top row.

*Table 3.* Quantitative comparison of simultaneous multi-task generation performance. We report DINOv2 and CLIP scores to assess object fidelity, and Success Rate to evaluate instruction adherence (i.e., successfully detecting all $K$ requested subjects).

| Method | DINO ↑ | CLIP ↑ | Success Rate ↑ |
|---|---|---|---|
| Average | 0.3971 | 0.6387 | 0.74 |
| Task Arithmetic | 0.4052 | 0.6455 | 0.76 |
| TIES | 0.4475 | 0.6498 | 0.69 |
| DARE | 0.5050 | 0.6485 | 0.62 |
| **SSR (Ours)** | **0.5704** | **0.7357** | **0.91** |

the same baselines used in RQ1.

**Metrics.** We employ a detection-based pipeline using Grounding DINO (Liu et al., 2024) to evaluate multi-subject generation. Specifically, we query Grounding DINO for the $K$ target subjects and select the bounding box with the highest confidence score for each subject. Subsequently, we compute CLIP and DINOv2 similarity between the cropped regions and ground truth references. To strictly penalize instruction neglect, undetected objects are assigned a similarity score of zero. Finally, we report the Success Rate, defined as the percentage of samples where all $K$ target objects are successfully detected.

**Quantitative Results.** As detailed in Table 3, SSR establishes a new state-of-the-art in multi-task execution. Regarding generation fidelity, SSR achieves a DINOv2 score of 0.5704 and a CLIP score of 0.7357. Compared to the strongest sparse baseline DARE, SSR yields an absolute improvement of 6.54% in DINOv2 and 8.72% in CLIP. More importantly, SSR demonstrates exceptional robustness in

instruction adherence with a 91% Success Rate. Baselines like TIES and DARE rely on parameter sparsification to mitigate conflicts; while this maintains reasonable fidelity, it leads to significant task loss, dropping Success Rates to 69% and 62%. Consequently, SSR avoids this suppression and outperforms DARE by 29% in this metric.

**Qualitative Results.** As shown in Figure 4, baselines exhibit characteristic failure modes under multi-task interference. In the dual-task case (top row), Linear Average suffers from style collapse, reverting to a cartoon domain. Task Arithmetic and TIES lose subject identity, generating generic orange or tabby cats instead of the specific grey target. DARE exhibits severe structural instability, erroneously hallucinating a second cat. In the three-object scenario (bottom row), while baselines manage to generate the object classes, they fail to preserve fine-grained visual traits; for instance, the specific features of the target bear are homogenized into a generic plushie. In contrast, SSR accurately reconstructs all requested subjects with their unique high-fidelity details and correct spatial arrangement.

### 4.3. RQ3: Generalization to Image Editing Tasks

**Objective and Protocol.** This section investigates the third research question: *Is the proposed method effective across different types of diffusion tasks, such as image editing?* To answer this, we design a dense multi-attribute editing benchmark focusing on precision facial retouching. We select three distinct editing tasks—lipstick, blush, and eyeshadow application—to evaluate the model's ability to resolve fine-grained parameter conflicts. The experiments are conducted on the FFHQ dataset (Karras et al., 2019), which is parti-

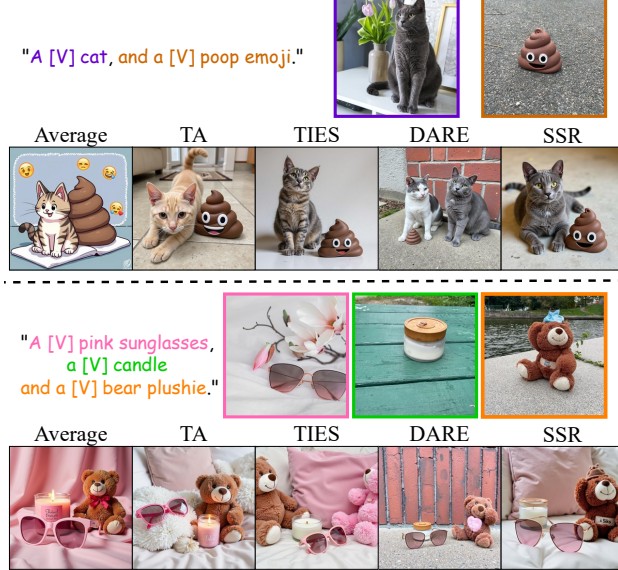

*Figure 4.* Visual comparison of simultaneous multi-task execution. The figure presents two experimental settings with different task counts: $K = 2$ (top panel) and $K = 3$ (bottom panel). For each setting, the composite prompt and the corresponding ground truth reference images are displayed above, followed by the generated results from different merging methods.

tioned into 400 images for training and 100 diverse images for testing.

**Implementation.** We employ commercial-grade image processing software to construct the training data. For each image in the training split, we synthesize three distinct target images, each corresponding to a single editing instruction. We then train three independent task-specific LoRAs on these source-target pairs. During the inference phase, these three LoRAs are merged to apply all makeup effects simultaneously on the test set. To establish a rigorous ground truth for evaluation, we apply the three editing instructions sequentially to the test images using the same commercial software. This serial execution produces high-quality compositional results, serving as the reference upper bound for evaluating the parallel merging performance. Consistent with the previous sections, we benchmark SSR against the same set of baselines used in RQ1.

**Metrics.** Quantitative performance is assessed using two key metrics: ArcFace similarity (Deng et al., 2019) is computed to verify identity preservation, ensuring the subject's facial features remain unaltered, while CLIP scores are calculated to measure the semantic alignment between the merged output and the sequentially edited ground truth.

**Quantitative Results.** As presented in Table 4, SSR consistently outperforms all baselines in both identity preservation and editing fidelity. Specifically, our method improves the ArcFace score by 1.39% compared to the strongest base-

*Table 4.* Quantitative comparison on the facial editing benchmark. The table reports the average ArcFace similarity and CLIP scores across the test set for SSR and baseline methods.

| Method | ArcFace Score ↑ | CLIP Score ↑ |
|---|---|---|
| Task Arithmetic | 0.9089 | 0.9348 |
| TIES-Merging | 0.9430 | 0.9529 |
| DARE | 0.9471 | 0.9464 |
| **SSR (Ours)** | **0.9610** | **0.9625** |

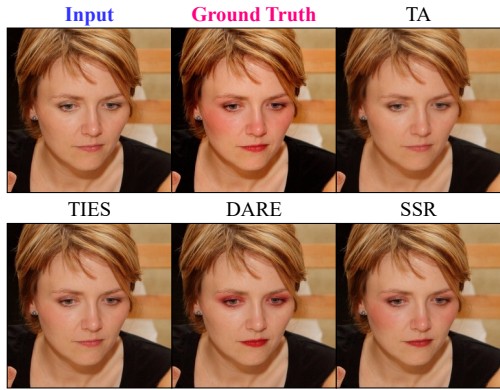

*Figure 5.* Visual comparison on the facial editing benchmark. We apply three makeup attributes (lipstick, blush, and eyeshadow) simultaneously. The figure displays the source image, the sequentially edited ground truth, and the generation results from different merging methods.

line DARE. Furthermore, it surpasses TIES-Merging by 0.96% in CLIP score, confirming that SSR effectively resolves parameter conflicts in dense editing tasks without compromising subject identity.

**Qualitative Results.** The visual comparison in Figure 5 further validates these findings. Task Arithmetic fails to inject the editing concepts, producing an output almost identical to the source. TIES-Merging exhibits very weak editing effects, resulting in washed-out colors. DARE suffers from severe attribute imbalance due to its sparsification and rescaling mechanism; it tends to over-amplify certain features (e.g., lipstick) while erroneously masking others. In contrast, SSR faithfully reconstructs all three target attributes—lipstick, blush, and eyeshadow—with balanced intensity and precise localization, closely matching the serial execution ground truth.

## 5. Conclusion

We introduce SSR, which recasts LoRA merging as signal routing in a unified low-rank subspace rather than parameter-space arithmetic, with a closed-form router that is provably optimal in the least-squares sense. Experiments confirm that SSR consistently outperforms state-of-the-art baselines.

## Limitations

SSR optimizes a local linear reconstruction objective, which does not theoretically guarantee global optimality in the full nonlinear diffusion process. Although our experiments show that this local optimum closely approximates the upper bound, the gap may widen under more extreme conditions. Additionally, when merging tasks with severe domain conflicts or high semantic overlap, stronger parameter interference makes routing more challenging, and performance may degrade. Finally, the ability to compose multiple concepts with high fidelity could potentially be misused for generating deceptive content, and we encourage responsible use of such techniques.

## Acknowledgements

This work was supported in part by the Shanghai Municipal Commission of Economy and Informatization, under Grant No. 2024-GZL-RGZN-01008. This work was also supported in part by the National Natural Science Foundation of China, under Grant Nos. 62192783, 62276128, and 62406140; the Young Elite Scientists Sponsorship Program by China Association for Science and Technology, under Grant No. 2023QNRC001; the Key Research and Development Program of Jiangsu Province, under Grant No. BE2023019; and the Jiangsu Natural Science Foundation, under Grant Nos. BK20221441 and BK20241200. The authors would like to thank the support of Huawei Ascend Cloud Ecological Development Project.

## Impact Statement

This paper presents work whose goal is to advance the field of machine learning. There are many potential societal consequences of our work, none of which we feel must be specifically highlighted here.

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

## Appendix Table of Contents

## A. Detailed Calculation Protocol for Activation Alignment (Fig. 1)

To rigorously quantify parameter interference (Figure 1), we visualize the activation alignment between task-specific prompts and their corresponding LoRA modules.

### A.1. Experimental Setup and Task Definitions

We selected 10 distinct concepts from the DreamBooth dataset to represent diverse tasks ($T_1 \dots T_{10}$). For each task, a dedicated LoRA module ($L_1 \dots L_{10}$) was trained on FLUX.1-dev. The target concepts include: *Bear Plushie, Candle, Cat, Colorful Sneaker, Dog, Pink Sunglasses, Poop Emoji, RC Car, Red Cartoon,* and *Teapot*. Each task is triggered by a specific prompt (e.g., "a sks [class]").

### A.2. Measurement Methodology

To ensure a fair and mathematically rigorous comparison, we adopt a unified definition of contribution. For both the baseline and our method, the total model update $\Delta h$ is linearly decomposable into task-specific components: $\Delta h = \sum_k h_k$. We quantify the Activation Energy for module $L_k$ under prompt $T_i$ by measuring the magnitude of its specific update vector $h_k$ before summation:

$$\text{Energy}(T_i, L_k) = \frac{1}{M} \sum_{m=1}^{M} \|h_k^{(m)}\|_2. \tag{16}$$

Although the mechanisms for generating $h_k$ differ, they represent the equivalent additive term in the final output equation:

- **Static Baseline (DARE):** We employ DARE with a sparsity rate of $p = 0.9$. To strictly replicate the physical merge where weights are summed ($\Delta W = \sum \lambda_k \tilde{W}_k$), we compute the contribution of the $k$-th branch as $h_k = \lambda_k B_k (M_k \odot A_k) x$. Here, $\lambda_k = 10$ is the rescaling factor. The sum of these individual $h_k$ vectors mathematically equals the output of the actual merged weight.

- **SSR (Ours):** The contribution is derived from the routed subspace signal. The input is first projected into the unified space $z = \mathbf{A}_{\text{comb}} x$ and transformed by the router $s = Rz$. We then slice the decorrelated signal $s$ to extract the component corresponding to task $k$. The specific contribution is $h_k = B_k s_{[k]}$, where $s_{[k]}$ denotes the signal slice precisely directed to the up-projection $B_k$. Since $\Delta h = \mathbf{B}_{\text{comb}} s = \sum B_k s_{[k]}$, this definition is structurally equivalent to the baseline's additive form.

### A.3. Visualization Protocol

To normalize varying feature magnitudes, we apply row-wise max-normalization: $\hat{E}_{i,k} = E_{i,k} / \max_j (E_{i,j})$. In the heatmap, diagonal elements ($\hat{E}_{i,i}$) ideally equal $1.0$. High off-diagonal values in the baseline indicate severe *crosstalk*, where unrelated modules are spuriously activated by task-specific instructions.

**Dataset for RQ1 and RQ2:**

**Dataset for RQ3:**

*Figure 6.* **Overview of the datasets used in our experiments.** The top panel displays the 10 custom subjects curated for the single-task (RQ1) and multi-task (RQ2) benchmarks. The bottom panel illustrates the three specific facial editing instructions designed for the RQ3 benchmark on the FFHQ dataset.

## B. Dataset and Baseline Details

### B.1. Dataset Construction

**Single-Task & Multi-Task Benchmark (RQ1 & RQ2).** We curate a dataset of 10 distinct subjects to evaluate concept preservation and composition. As illustrated in the top panel of Figure 6, the subjects include: *bear plushie, candle, cat, colorful sneaker, dog, poop emoji, pink sunglasses, red cartoon, rc car,* and *tea pot.* These subjects were selected to cover a diverse range of categories (animals, toys, objects) and texture complexities.

**Facial Editing Benchmark (RQ3).** For fine-grained editing tasks, we utilize the FFHQ dataset (Karras et al., 2019). We define three specific editing instructions as shown in the bottom panel of Figure 6:

- **Lipstick:** "Change the lipstick to a pomegranate red style"

- **Blush:** "Apply dragon fruit style blush"

- **Eyeshadow:** "Apply lush style eyeshadow"

### B.2. Baselines

We benchmark our method against established parameter merging paradigms. A critical implementation detail in our experiments is that we perform all merging operations on the reconstructed weight updates $\Delta W = BA \in \mathbb{R}^{d \times d}$, rather than on the individual low-rank factors $A$ and $B$. We empirically observed that merging factorized matrices directly leads to performance degradation, as the latent bases of independently trained LoRAs are not spatially aligned.

Therefore, for a set of $K$ tasks, we define the task vector $\tau_k$ for the $k$-th task as its LoRA update:

$$\tau_k = \Delta W_k = B_k A_k. \tag{17}$$

**Linear Average.** A naive approach to multi-task merging is to average the parameters of all models. This assumes that the optimal solution lies at the centroid of the task-specific solutions:

$$\Delta W_{\text{Avg}} = \frac{1}{K} \sum_{k=1}^{K} \tau_k. \tag{18}$$

While simple, averaging often leads to "concept dilution," where the magnitude of task-specific features is reduced by a factor of $K$, diminishing the model's ability to trigger specific concepts.

**Task Arithmetic.** Task Arithmetic (Ilharco et al., 2023) treats model editing as vector operations in the weight space. It computes the unified update by summing the task vectors, often controlled by a global scaling factor $\lambda$:

$$\Delta W_{\text{TA}} = \lambda \sum_{k=1}^{K} \tau_k. \tag{19}$$

**Relation to SSR.** Ideally, if we set the routing matrix in our framework to the identity matrix, i.e., $R = \mathbf{I}_{Kr} \in \mathbb{R}^{Kr \times Kr}$, the output of our model becomes:

$$\Delta W_{\text{SSR}} = \mathbf{B}_{\text{comb}} \mathbf{I} \mathbf{A}_{\text{comb}} = \begin{bmatrix} B_1 & \dots & B_K \end{bmatrix} \begin{bmatrix} A_1 \\ \vdots \\ A_K \end{bmatrix} = \sum_{k=1}^{K} B_k A_k. \tag{20}$$

This derivation reveals that **Task Arithmetic (with $\lambda = 1$) is mathematically equivalent to a special case of SSR where $R = \mathbf{I}$**. It corresponds to a naive concatenation strategy in the rank dimension without any cross-task signal regulation. This explains why Task Arithmetic is susceptible to destructive interference: it blindly superimposes signal directions without orthogonalizing them.

**TIES-Merging.** TIES-Merging (Yadav et al., 2023) addresses interference by resolving sign conflicts and pruning redundant parameters. It operates on the flattened task vectors through a three-step pipeline: Trim, Elect Sign, and Merge.

$$\Delta W_{\text{TIES}} = \text{Mean}\left(\text{SignSelect}\left(\text{Top-}k\left(\{\tau_1, \dots, \tau_K\}\right)\right)\right). \tag{21}$$

By keeping only the most significant parameters and enforcing direction consensus, TIES aims to reduce the "noise" introduced by conflicting updates.

**DARE (Drop And REscale).** DARE (Yu et al., 2024) employs a stochastic approach to sparsification. It randomly drops parameters from each task vector $\tau_k$ with a probability $1-p$ and rescales the remaining parameters to maintain the magnitude expectation:

$$\Delta W_{\text{DARE}} = \sum_{k=1}^{K} \left( \frac{M_k \odot \tau_k}{p} \right), \quad M_k \sim \text{Bernoulli}(p). \tag{22}$$

This method relies on the hypothesis that the delta weights are highly redundant and that random sparsification can preserve the core function of the model while vacating space for other tasks.

**RobustMerge.** RobustMerge (Zeng et al., 2025) is a training-free merging method tailored to parameter-efficient modules. Its central observation is that, under low-rank decomposition, the dominant singular directions of $\tau_k$ are sensitive to interference, so preserving the *direction* of each task vector is more important than preserving its magnitude. To this end, RobustMerge (i) prunes parameters and rescales coefficients using inter-parameter relations on the singular values, stabilizing the principal directions away from task interference, and (ii) applies a cross-task normalization step to balance the contributions of different tasks. Compared with sign- or sparsity-based heuristics such as TIES and DARE, RobustMerge focuses explicitly on direction robustness in the low-rank subspace.

**IterIS.** IterIS (Chen et al., 2025a) formulates LoRA merging as a layer-wise optimization problem and refines its solution through an iterative inference-solving loop. Given a small set of unlabeled calibration samples, IterIS alternates between (i) running the current merged model to obtain updated layer-wise activations, and (ii) solving a regularized least-squares problem to minimize the discrepancy between the merged and per-task outputs. An adaptive task weighting is further introduced to mitigate imbalance across tasks. Compared with prior optimization-based merging baselines, IterIS reduces the required number of calibration samples to about 1–5% while converging in a few layer-wise iterations, making it a strong optimization-based competitor for LoRA merging.

*Table 5.* Single-task preservation of SSR on FLUX.1-dev when scaling to larger merge counts.

| Method | $K = 9$ | | $K = 12$ | | $K = 15$ | | $K = 18$ | | $K = 21$ | |
|---|---|---|---|---|---|---|---|---|---|---|
| | DINO | CLIP | DINO | CLIP | DINO | CLIP | DINO | CLIP | DINO | CLIP |
| Upper Bound | 0.744 | 0.845 | 0.744 | 0.845 | 0.744 | 0.845 | 0.744 | 0.845 | 0.744 | 0.845 |
| SSR | 0.671 | 0.785 | 0.652 | 0.774 | 0.630 | 0.751 | 0.604 | 0.737 | 0.573 | 0.714 |
| Recovery Rate | 90.2% | 92.9% | 87.6% | 91.6% | 84.7% | 88.9% | 81.2% | 87.2% | 77.0% | 84.5% |

*Table 6.* Real-world community LoRA composition results.

| Method | CLIP Score |
|---|---|
| Average | 0.652 |
| Task Arithmetic | 0.684 |
| TIES | 0.735 |
| DARE | 0.712 |
| RobustMerge | 0.768 |
| **SSR (Ours)** | **0.821** |

## C. Scalability and Real-World Composition

**Scaling to larger merge counts.** To further evaluate scalability, we extend the FLUX.1-dev single-task preservation benchmark to larger merge counts up to $K = 21$. As shown in Table 5, SSR maintains strong recovery rates as the number of merged LoRAs increases, although parameter interference naturally becomes more severe at larger $K$.

**Real-world community LoRA composition.** We also evaluate a practical composition setting using three community LoRAs downloaded from the Liblib platform, covering lighting, portrait beautification, and portrait stylization. We use serial execution of the three LoRAs as the reference and report CLIP similarity in Table 6. SSR achieves the highest score among all evaluated merging methods.

## D. Generalization Beyond Diffusion

Although our main focus is diffusion models, we further validate SSR on non-generative downstream tasks using the GLUE benchmark. Following the DOGE TA setting (Wei et al., 2025), we compare SSR with representative model-merging baselines across eight GLUE tasks, including Concrete TA (Tang et al., 2023). As shown in Table 7, SSR achieves the highest average score, indicating that the routing-based merging mechanism can also transfer beyond diffusion-based synthesis.

## E. Analysis of RegMean in High-Dimensional Diffusion Settings

In this section, we provide a detailed analysis of the RegMean baseline (Jin et al., 2023). While RegMean offers a theoretically closed-form solution for weight merging, its application to high-dimensional Diffusion Transformers (such as FLUX.1) presents fundamental difficulties regarding numerical stability and computational feasibility.

**Mathematical Formulation.** RegMean minimizes the $L_2$ distance between the activations of the merged model and the individual models. The optimal merged weight $W_M$ is calculated as:

$$W_M = \left( \sum_{i \in \mathcal{K}} X_i^T X_i \right)^{-1} \sum_{i \in \mathcal{K}} \left( X_i^T X_i W_i \right), \tag{23}$$

where $X_i \in \mathbb{R}^{N \times d}$ represents the input feature matrix for the $i$-th task, and $W_i$ is the corresponding task-specific weight. The term $\sum X_i^T X_i$ represents the global covariance (Gram) matrix of the input activations.

**Inherent Singularity and Contrast with SSR.** The fundamental limitation of RegMean in this setting lies in the dimensionality of the inversion problem.

*Table 7.* Generalization to non-generative downstream tasks on GLUE.

| Method | CoLA | MNLI | MRPC | QNLI | QQP | RTE | SST2 | STSB | Avg. |
|---|---|---|---|---|---|---|---|---|---|
| Weight Averaging | 69.7 | 59.7 | 78.9 | 90.1 | 83.8 | 80.5 | 91.2 | 72.0 | 78.2 |
| Task Arithmetic | 68.8 | 55.2 | 78.7 | 89.8 | 83.7 | 79.1 | 91.5 | 72.4 | 77.4 |
| Ties Merging | 68.3 | 56.3 | 79.4 | 89.8 | 83.7 | 79.4 | 91.6 | 71.2 | 77.5 |
| Concrete TA | 69.1 | 58.1 | 78.4 | 89.9 | 83.5 | 79.4 | 91.6 | 73.4 | 78.0 |
| DOGE TA | 69.1 | 71.9 | 80.9 | 90.3 | 83.5 | 79.8 | 92.5 | 71.1 | 79.9 |
| **SSR** | **69.3** | **73.3** | **82.1** | 90.1 | 83.6 | **81.4** | **92.7** | **74.9** | **80.9** |

- **RegMean (Global Space):** Requires inverting the global covariance matrix of size $d \times d$. In FLUX.1, $d$ is substantial (e.g., $d \approx 12,288$). Under a one-shot calibration setting ($N \approx 4,096$), we face a regime where $N \ll d$. This renders the matrix inherently rank-deficient and mathematically singular, making direct inversion impossible.

- **SSR (Subspace):** In contrast, our method projects signals into a compact subspace of rank $Kr$ (e.g., $Kr \approx 96$) *before* computing statistics. Since $N \gg Kr$, the resulting subspace correlation matrix is well-conditioned and naturally invertible without requiring aggressive regularization.

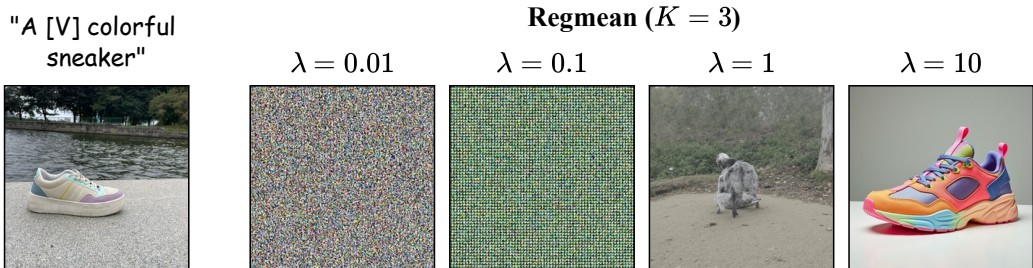

*Figure 7.* **Visual analysis of RegMean stability.** Since the RegMean covariance matrix is singular ($d \gg N$), we introduce regularization $\lambda\mathbf{I}$ to enable inversion. However, the results reveal a severe trade-off due to the ill-conditioned nature of the matrix: small $\lambda$ fails to suppress numerical explosion (noise), while large $\lambda$ dominates the signal, erasing task-specific features (identity loss).

**The Regularization Dilemma.** Since the covariance matrix is singular, introducing a Tikhonov regularization term $\lambda\mathbf{I}$ is mathematical requisite to obtain a computable solution for RegMean:

$$W_M \approx \left( \sum_{i \in \mathcal{K}} X_i^T X_i + \lambda\mathbf{I} \right)^{-1} \sum_{i \in \mathcal{K}} \left( X_i^T X_i W_i \right). \tag{24}$$

However, we find that due to the severe ill-conditioning of the high-dimensional covariance matrix, this workaround fails to yield a valid operating point (as shown in Figure 7):

- **Numerical Instability ($\lambda \leq 0.1$):** When $\lambda$ is small, it is insufficient to correct the condition number. The inversion is dominated by numerical errors from near-zero eigenvalues, resulting in chaotic, high-frequency artifacts (Columns 1-2).

- **Identity Dilution ($\lambda \geq 1$):** Increasing $\lambda$ makes the matrix numerically invertible, but the regularization term begins to dominate the covariance sum. This effectively "washes out" the task-specific correlations ($X_i^T X_i$), leading to the loss of subject identity (Columns 3-4).

# F. Additional Results on Qwen-Image

To further validate the architectural universality of our method, we provide additional quantitative and qualitative comparisons using the Qwen-Image backbone (Wu et al., 2025). Qwen-Image exhibits different feature space characteristics from FLUX.1; nevertheless, the interference patterns of standard merging methods remain consistent.

*Table 8.* Quantitative evaluation of single-task capability preservation on Qwen-Image under multi-task interference. We report the average DINOv2 and CLIP scores across varying numbers of merged tasks ($K$). The Upper Bound (shown in gray) represents the performance of a standalone LoRA without merging.

| Method | K=3 | | K=5 | | K=7 | | K=9 | |
|---|---|---|---|---|---|---|---|---|
| | DINO | CLIP | DINO | CLIP | DINO | CLIP | DINO | CLIP |
| Average | 0.5443 | 0.7624 | 0.4899 | 0.7221 | 0.4531 | 0.7103 | 0.4467 | 0.6913 |
| Task Arithmetic | 0.6894 | 0.8036 | 0.5861 | 0.7872 | 0.5822 | 0.7816 | 0.4772 | 0.7453 |
| TIES | 0.7455 | 0.8427 | 0.6292 | 0.8205 | 0.5971 | 0.8147 | 0.5716 | 0.7902 |
| DARE | 0.7308 | 0.8372 | 0.5996 | 0.8109 | 0.5975 | 0.7975 | 0.5557 | 0.7768 |
| **SSR (Ours)** | **0.7626** | **0.8761** | **0.7571** | **0.8746** | **0.7540** | **0.8673** | **0.7461** | **0.8654** |
| *Recovery Rate* | 99.2% | 99.7% | 98.4% | 99.5% | 98.0% | 98.7% | 97.0% | 98.5% |
| *Upper Bound* | *0.7691* | *0.8786* | *0.7691* | *0.8786* | *0.7691* | *0.8786* | *0.7691* | *0.8786* |

**Quantitative Results.** Table 8 reports the evaluation results on Qwen-Image under the same variable-scale merging protocol as in the main paper. In the high-interference setting ($K = 9$), SSR surpasses the strongest baseline TIES by 17.45% in DINOv2 similarity. Moreover, baselines degrade rapidly as task counts increase: the TIES score drops by 17.39% from $K = 3$ to $K = 9$, whereas SSR exhibits a decline of only 1.65% under identical conditions. In terms of fidelity retention, SSR consistently recovers over 97.0% of the single-task oracle performance on Qwen-Image.

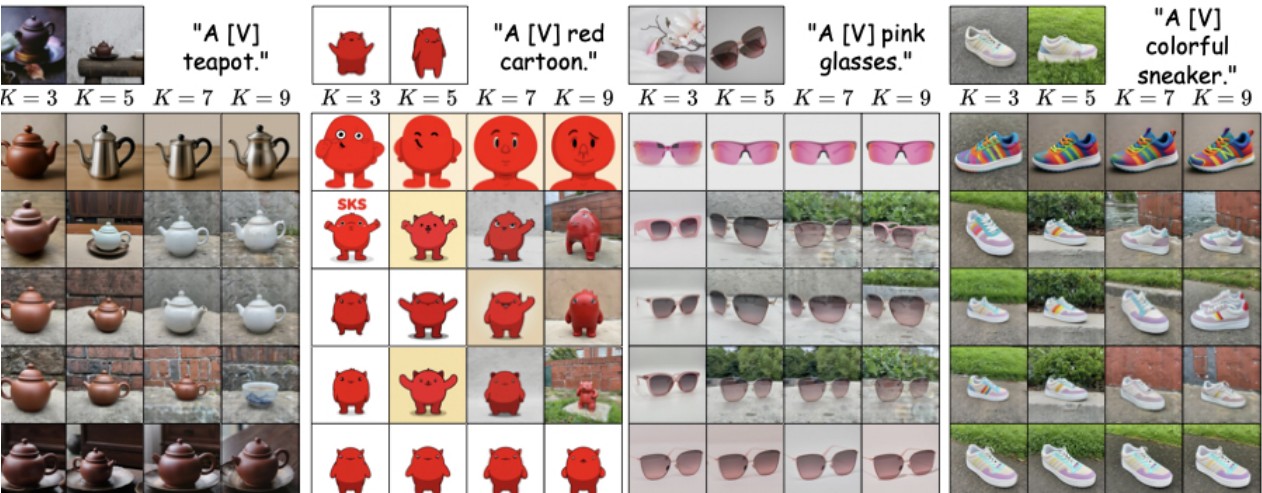

*Figure 8.* **Visual comparison of single-task preservation on Qwen-Image .** We illustrate the generation quality of the target subject mixed with increasing numbers of distractor LoRAs ($K \in \{3, 5, 7, 9\}$). While baseline methods (e.g., Task Arithmetic, DARE) exhibit severe semantic collapse and identity loss—often generating unrecognizable noise or generic concepts under high interference ($N = 9$)—SSR consistently preserves the structural and textural details of the subject. This confirms that the subspace routing mechanism is robust across different transformer-based diffusion architectures.

**Qualitative Results.** Figure 8 presents the visual results. Consistent with our findings on FLUX.1, dense merging methods (Linear Average, Task Arithmetic) suffer from rapid concept dilution. As the number of tasks increases to $K = 7$ and $K = 9$, these methods fail to retain the target subject's defining traits, resulting in generic or corrupted outputs. Sparse methods like TIES and DARE, while occasionally preserving outlines, struggle with texture consistency and often introduce high-frequency artifacts or erroneous attribute bindings due to the mismatch in the Qwen-Image feature space.

In sharp contrast, SSR demonstrates exceptional stability. Even in the most challenging regime ($K = 9$), SSR successfully disentangles the target signal from the interference, generating images that are semantically aligned with the single-task Oracle. This visual evidence corroborates the quantitative results reported in Table 8, where SSR surpasses the strongest baseline by over 34% in DINOv2 similarity on the Qwen-Image.

# G. Calibration Robustness

## G.1. Ablation Study on Calibration Steps

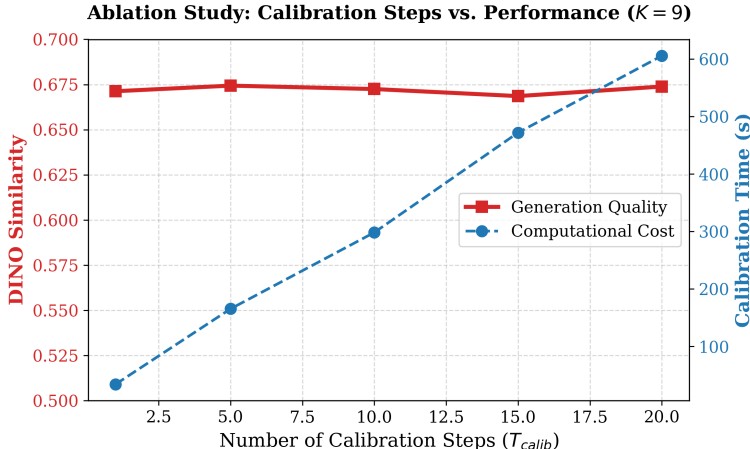

*Figure 9.* **Impact of calibration steps on performance and cost** ($K = 9$)**.** The red solid line indicates the generation fidelity (DINO Similarity), while the blue dashed line represents the calibration time cost. We observe that model performance saturates immediately at $T = 1$, whereas the computational overhead increases linearly. This justifies our choice of one-shot calibration as the optimal efficiency-performance trade-off.

To validate the efficiency of our one-shot calibration strategy, we conduct an ablation study on the number of calibration steps ($T_{calib}$) required to estimate the subspace routing matrix. We evaluate the performance on the FLUX.1 backbone under the most computationally demanding setting with $K = 9$ tasks, varying $T_{calib} \in \{1, 5, 10, 15, 20\}$.

**Empirical Observations.** As illustrated in Figure 9, increasing the number of calibration steps yields negligible gains in generation quality. The DINO similarity score remains statistically stable within a narrow range (e.g., 0.6713 at $T = 1$ versus 0.6739 at $T = 20$), indicating that the subspace statistics converge rapidly. In contrast, the time cost scales linearly with $T_{calib}$. Since SSR requires performing inference for each of the $K$ tasks during the calibration phase, a 20-step calibration for 9 tasks incurs a substantial overhead (>600s), whereas our proposed **one-shot calibration** ($T = 1$) completes in just 34.26 seconds. Consequently, $T = 1$ provides the most efficient solution without compromising fidelity.

**Mechanism Analysis: Decoupling Routing from Generation.** The sufficiency of one-shot calibration stems from a fundamental design principle: SSR decouples the temporal generation capability from the signal conflict resolution.

The complex, time-dependent generative dynamics are entirely preserved within the original LoRA matrices (conceptually $A$ and $B$). Since these matrices are already trained to handle inputs across all diffusion timesteps, the merged model inherently inherits this temporal generalization. Consequently, our router $\mathbf{R}$ is not required to "learn" or approximate the specific operations for each timestep. Instead, its sole function is to redistribute the signal flow within the task subspace to ensure orthogonality. Because the geometric orientation of the task-specific subspaces remains structurally stable, a routing matrix $\mathbf{R}$ derived from a single representative timestep is sufficient to globally resolve parameter conflicts, without the need to track the entire diffusion trajectory.

## G.2. Prompt Template Stability

We further evaluate whether one-shot calibration is sensitive to prompt wording. At $K = 3$, we test five calibration prompt templates: **T1**: `A high quality photo of a [V] [obj]`; **T2**: `A professional studio shot of a [V] [obj] against a neutral solid background`; **T3**: `A detailed artistic illustration depicting a [V] [obj]`; **T4**: `A [V] [obj] placed on a clean wooden surface in a well lit room`; and **T5**: `A [V] [obj] captured with dramatic cinematic lighting and high contrast`. As shown in Table 9, the cross-template variance remains very small, indicating that the final merged model is stable to calibration prompt wording.

*Table 9.* Prompt template stability of one-shot calibration at $K = 3$.

| Metric | T1 | T2 | T3 | T4 | T5 | Mean | Var |
|---|---|---|---|---|---|---|---|
| Avg CLIP | 0.7915 | 0.8196 | 0.8160 | 0.8267 | 0.7983 | 0.8104 | 0.00067 |
| Avg DINO | 0.7405 | 0.7283 | 0.7324 | 0.7287 | 0.7260 | 0.7312 | 0.00035 |

*Table 10.* Stability under different random distractor sets and random seeds at $K = 3$.

| Condition | DINO Mean | DINO Std | CLIP Mean | CLIP Std |
|---|---|---|---|---|
| Random Distractor Sets | 0.729 | 0.011 | 0.810 | 0.008 |
| Random Seeds | 0.733 | 0.007 | 0.813 | 0.005 |

### G.3. Stability Across Randomness

We further evaluate the stability of SSR under different random distractor sets and random seeds. At $K = 3$, we report the mean and standard deviation across five independent runs in Table 10. The results remain stable under both sources of randomness. The worst-case drops are limited to 0.017 in DINO and 0.012 in CLIP, which indicates that the gains are not caused by a favorable distractor selection or seed choice.

## H. Theoretical Analysis of Finite-Sample Error

In Section 3.2, we constructed the Subspace Signal Router $R$ using empirical second-order statistics ($\hat{\mathbf{G}}$ and $\hat{\mathbf{Q}}$) derived from a calibration set of size $N$. All experiments use the unregularized OLS form $R = \hat{\mathbf{Q}}\hat{\mathbf{G}}^{-1}$. In this section, we provide a rigorous bound on the estimation error under the same formulation. We prove that due to the low-dimensional nature of the LoRA subspace, our constructed router converges rapidly to the population optimal router $R^*$ as $N$ increases.

### H.1. Setup and Notations

Let the unified subspace dimension be $D_{\text{sub}} = K \cdot r$. We define the router operates in this compact space $\mathbb{R}^{D_{\text{sub}} \times D_{\text{sub}}}$. Let $\mathcal{D}$ denote the underlying distribution of the projected activations $z \in \mathbb{R}^{Kr}$ and task-specific target signals $y \in \mathbb{R}^{Kr}$.

The population optimal router (defined by expected statistics over the true distribution) is:

$$R^* = \mathbf{Q}^*(\mathbf{G}^*)^{-1}, \tag{25}$$

where $\mathbf{G}^* = \mathbb{E}[zz^\top] \in \mathbb{R}^{Kr \times Kr}$ and $\mathbf{Q}^* = \mathbb{E}[yz^\top] \in \mathbb{R}^{Kr \times Kr}$.

The empirical router (constructed in our method using $N$ samples) is:

$$\hat{R} = \hat{\mathbf{Q}}\hat{\mathbf{G}}^{-1}, \tag{26}$$

where $\hat{\mathbf{G}} = \frac{1}{N}\sum_{i=1}^{N} z_i z_i^\top$ and $\hat{\mathbf{Q}} = \frac{1}{N}\sum_{i=1}^{N} y_i z_i^\top$.

Our objective is to derive the upper bound for the estimation error $\|\hat{R} - R^*\|_2$.

### H.2. Main Theorem

**Theorem H.1** (Sample Complexity in Low-Rank Subspace). *Assume the feature vectors $z$ are sub-Gaussian with parameter $\sigma^2$ and bounded norm. Let the population correlation matrix be well-conditioned, i.e., $\lambda_{\min}(\mathbf{G}^*) \geq \mu > 0$. For any $\delta \in (0, 1)$, provided the calibration sample size $N$ is sufficiently large, the difference between our constructed router $\hat{R}$ and the population optimum $R^*$ is bounded with probability at least $1 - \delta$ by:*

$$\|\hat{R} - R^*\|_2 \leq \frac{C}{\mu}\left(1 + \frac{1}{\mu}\right)\sqrt{\frac{Kr + \log(1/\delta)}{N}}, \tag{27}$$

*where $C$ is a constant depending on the sub-Gaussian norms of the data.*

## H.3. Proof of Theorem H.1

**1. Error Decomposition.** We decompose the error term $\Delta R = \hat{R} - R^*$ using the identity $A^{-1} - B^{-1} = A^{-1}(B - A)B^{-1}$. We have:

$$
\begin{aligned}
\hat{R} - R^* &= \hat{\mathbf{Q}}\hat{\mathbf{G}}^{-1} - \mathbf{Q}^*(\mathbf{G}^*)^{-1} \\
&= (\hat{\mathbf{Q}} - \mathbf{Q}^*)(\mathbf{G}^*)^{-1} + \hat{\mathbf{Q}}\left(\hat{\mathbf{G}}^{-1} - (\mathbf{G}^*)^{-1}\right) \\
&= \underbrace{(\hat{\mathbf{Q}} - \mathbf{Q}^*)(\mathbf{G}^*)^{-1}}_{\text{Term A}} + \underbrace{\hat{\mathbf{Q}}\hat{\mathbf{G}}^{-1}(\mathbf{G}^* - \hat{\mathbf{G}})(\mathbf{G}^*)^{-1}}_{\text{Term B}}.
\end{aligned}
\tag{28}
$$

**2. Bounding the Inverses.** By assumption, $\lambda_{\min}(\mathbf{G}^*) \geq \mu$, hence $\|(\mathbf{G}^*)^{-1}\|_2 \leq 1/\mu$. When the empirical covariance concentration satisfies $\|\hat{\mathbf{G}} - \mathbf{G}^*\|_2 \leq \mu/2$, Weyl's inequality gives $\lambda_{\min}(\hat{\mathbf{G}}) \geq \mu/2$. Thus, the spectral norms of the inverse matrices are bounded:

$$
\|(\mathbf{G}^*)^{-1}\|_2 \leq \frac{1}{\mu}, \quad \|\hat{\mathbf{G}}^{-1}\|_2 \leq \frac{2}{\mu}.
\tag{29}
$$

**3. Concentration of Statistics in Subspace.** This is the crucial step where the subspace dimensionality plays a role. We apply the Matrix Bernstein Inequality to bound the deviation of the empirical covariance matrix in the $Kr$-dimensional space. For a sub-Gaussian vector $z \in \mathbb{R}^{Kr}$, the convergence rate of the sample covariance $\hat{\mathbf{G}}$ to $\mathbf{G}^*$ is governed by the dimension $Kr$. With high probability $1 - \delta$:

$$
\|\hat{\mathbf{G}} - \mathbf{G}^*\|_2 \lesssim \sqrt{\frac{Kr + \log(1/\delta)}{N}}.
\tag{30}
$$

Similarly for the cross-covariance $\mathbf{Q}$:

$$
\|\hat{\mathbf{Q}} - \mathbf{Q}^*\|_2 \lesssim \sqrt{\frac{Kr + \log(1/\delta)}{N}}.
\tag{31}
$$

**4. Final Assembly.** Substituting the bounds from Step 2 and Step 3 into the decomposition in Step 1:

$$
\begin{aligned}
\|\hat{R} - R^*\|_2 &\leq \|\hat{\mathbf{Q}} - \mathbf{Q}^*\|_2 \frac{1}{\mu} + \|\hat{\mathbf{Q}}\|_2 \frac{2}{\mu}\|\mathbf{G}^* - \hat{\mathbf{G}}\|_2 \frac{1}{\mu} \\
&\leq \frac{1}{\mu}\mathcal{O}\left(\sqrt{\frac{Kr}{N}}\right) + \frac{2K_Q}{\mu^2}\mathcal{O}\left(\sqrt{\frac{Kr}{N}}\right) \\
&= \mathcal{O}\left(\frac{1}{\mu}\left(1 + \frac{1}{\mu}\right)\sqrt{\frac{Kr}{N}}\right).
\end{aligned}
\tag{32}
$$

This completes the proof.

## H.4. Remark on Subspace Efficiency

The bound derived above highlights a fundamental advantage of performing routing within the LoRA subspace. The convergence rate is dependent on the term $\sqrt{Kr}$. Since our method operates in the bottleneck dimension where $Kr \ll d$ (e.g., $Kr \approx 64$ while the model width $d \approx 4096$), the numerator in the error bound is extremely small. This implies that our heuristic router $\hat{R}$ requires significantly fewer calibration samples $M$ to reliably approximate the optimal routing logic compared to methods operating in the full parameter space.

