# OpenReview forum: "SSR-Merge: Subspace Signal Routing for Training-Free LoRA Merging in Diffusion Models"
_ICML.cc/2026/Conference — ICML 2026 regular_

### Official Review · Reviewer_E1tF · 2026-02-24

**Soundness:** 1
**Presentation:** 2
**Significance:** 1
**Originality:** 2
**Overall Recommendation:** 2
**Confidence:** 4

**Summary:**

The paper tackles the interference issues that arise when merging multiple LoRA modules into a diffusion model. Standard averaging methods often fail here because conflicting signals clash in the shared parameter space. The authors propose SSR to handle this by treating merging as a signal routing problem. They construct a unified subspace and use a router to disentangle the signals, proving that this mathematically minimizes reconstruction error like an OLS estimator. I like that they use a streaming algorithm to keep memory usage low and eventually absorb the router into the weights to avoid latency. Results show it outperforms established baselines like TIES on various generative tasks.

**Compliance With Llm Reviewing Policy:**

Affirmed.

**Final Justification:**

I maintain to Reject. While I appreciate authors' effort in providing additional LLM results and complexity analysis, rebuttal reinforces my concern that the method's "optimality" is fundamentally flawed:

the rank grows linearly with the number of tasks, which contradicts the lightweight philosophy of LoRA and invalidates the claimed OLS benefits the moment post-hoc SVD compression is applied to regain efficiency.

SSR essentially achieves better results by expanding parameter budget ($Kr$) rather than solving interference within a fixed capacity, and I remain skeptical that a linear reconstruction loss can truly govern the highly non-linear generative dynamics of diffusion models.

Technical overhead and reliance on capacity expansion outweigh incremental performance gains shown in narrow evaluation settings.

**Key Questions For Authors:**

I struggle to see how this differs from a standard Mixture of Experts. The routing mechanism feels identical to existing gating techniques so I need you to clarify the distinct novelty here. Since you concatenate LoRAs the rank grows with every task which raises serious scalability concerns that are not addressed. Why are there no experiments on language models given that LoRA is standard there? I am also skeptical that calibrating the router on a single time step is sufficient for the entire diffusion process. Finally the comparison against baselines like TIES seems unfair because you use a much larger rank budget. I suspect the baselines might match your performance if allowed the same capacity.

**Limitations:**

The manuscript completely omits a discussion on limitations. The biggest issue is that concatenating LoRAs increases rank linearly which creates a massive scalability bottleneck for large task counts. I also doubt a simple linear router is optimal for the highly non linear diffusion process. The scope is too narrow as they ignore LLMs entirely. Finally the one shot calibration likely becomes unstable depending on the specific prompts used.

**Strengths And Weaknesses:**

Strengths
The paper tackles a valid issue with parameter interference in diffusion models and the engineering effort to keep inference latency zero is commendable. I also admit the visual results for subject preservation look better than simple averaging.

Weaknesses
The novelty is overstated as the subspace routing concept is essentially just a repackaged Mixture of Experts approach. By concatenating LoRAs along the rank dimension you are not really merging parameters but simply expanding the model which creates scalability issues that the paper ignores. The evaluation is also far too narrow relying on a handful of disjoint tasks and facial attributes without any LLM experiments. Finally proving OLS optimality for feature reconstruction does not guarantee better generation quality in practice.

---

> ### Author Rebuttal · Authors · 2026-03-31
>
> > **W1 & Q1: Distinction from Mixture of Experts**
>
> We respectfully disagree that SSR is a repackaged MoE. Although both use the word "routing", the two methods differ fundamentally, as listed below.
>
> | | MoE | SSR |
> | :--- | :--- | :--- |
> | Category | Model architecture | Model merging method |
> | Routing | Learned gating, input-dependent | OLS regression, computed once |
> | Operates on | Activations at every forward pass | Weights as one-time construction |
> | Output | Multi-branch with persistent experts | Single merged weight matrix |
>
> MoE dynamically selects which expert processes each token during inference. SSR combines $K$ separate LoRA matrices into one merged adapter with no experts, no gating, and no conditional computation at inference. The word "routing" appears in both but refers to completely different operations.
>
> > **W2 & Q2 & Q5 & L1: Rank budget fairness and scalability**
>
> We clarify two facts proving the comparison is fair and scalable.
>
> **Fair comparison.** All baselines share the exact same capacity as SSR. As shown in Eq. 19 in Appendix B.2 of the paper, arithmetic summation of LoRA updates is mathematically identical to rank-dimension concatenation:
> $$\\sum_{k=1}^K B_k A_k = \\begin{bmatrix} B_1 & \\dots & B_K \\end{bmatrix} \\begin{bmatrix} A_1 \\\\ \\vdots \\\\ A_K \\end{bmatrix}$$
> Baselines like TIES inherently operate within the same expanded subspace — our method uses no larger rank budget.
>
> **Scalability.** Reducing rank is trivial via SVD. As shown below, compressing from $Kr$ to $r$ causes only marginal degradation, confirming gains stem from optimal routing rather than parameter expansion.
>
> | Method | $K$=3 DINO | $K$=3 CLIP | $K$=5 DINO | $K$=5 CLIP | $K$=7 DINO | $K$=7 CLIP | $K$=9 DINO | $K$=9 CLIP |
> | :--- | :--- | :--- | :--- | :--- | :--- | :--- | :--- | :--- |
> | SSR Original | 0.7342 | 0.8144 | 0.7059 | 0.7951 | 0.6868 | 0.7798 | 0.6713 | 0.7850 |
> | SSR SVD Compressed | 0.7325 | 0.8152 | 0.7021 | 0.7928 | 0.6795 | 0.7712 | 0.6582 | 0.7715 |
>
> > **W3 & Q3 & L3: Generalization to language models**
>
> Our scope targets parameter interference in visual generation. To demonstrate generalizability, we conducted LLM experiments following the DOGE TA setting on the GLUE benchmark. SSR achieves the highest average score, confirming its effectiveness on language models.
>
> | Method | CoLA | MNLI | MRPC | QNLI | QQP | RTE | SST2 | STSB | Avg. |
> | :--- | :--- | :--- | :--- | :--- | :--- | :--- | :--- | :--- | :--- |
> | Weight Averaging | **69.7** | 59.7 | 78.9 | 90.1 | **83.8** | 80.5 | 91.2 | 72.0 | 78.2 |
> | Task Arithmetic | 68.8 | 55.2 | 78.7 | 89.8 | 83.7 | 79.1 | 91.5 | 72.4 | 77.4 |
> | Ties Merging | 68.3 | 56.3 | 79.4 | 89.8 | 83.7 | 79.4 | 91.6 | 71.2 | 77.5 |
> | Concrete TA | 69.1 | 58.1 | 78.4 | 89.9 | 83.5 | 79.4 | 91.6 | 73.4 | 78.0 |
> | DOGE TA | 69.1 | 71.9 | 80.9 | **90.3** | 83.5 | 79.8 | 92.5 | 71.1 | 79.9 |
> | SSR | 69.3 | **73.3** | **82.1** | 90.1 | 83.6 | **81.4** | **92.7** | **74.9** | **80.9** |
>
> > **W4 & L2: Relationship between OLS optimality and generation quality**
>
> We agree that local OLS optimality does not theoretically guarantee generation quality. We also compare against the global optimum in the paper, and Table 1 in Section 4.1 shows that SSR recovers over 90% of the upper bound across all settings, demonstrating that our local reconstruction optimality closely approximates the global optimum.
>
> > **Q4: Sufficiency of single-timestep calibration**
>
> Our ablation in Appendix E of the paper demonstrates that single-step calibration achieves performance equivalent to multi-step calibration, owing to our subspace design.
>
> > **L4: Prompt stability**
>
> We conducted prompt stability experiments showing variance across five diverse templates is below 0.1%. Please refer to our response to W2 & Q1 for Reviewer SBYq for details.

---

> > ### Author Rebuttal · Reviewer_E1tF · 2026-04-04
> >
> > I sincerely thank the authors for their rebuttal, but I respectfully maintain my score of 2.
> >
> > First, evaluating LLMs on GLUE is little bit outdated; modern validation requires generative or reasoning tasks. Second, SVD compression is merely a post-hoc patch. The merging phase still demands computing within a massively expanded Kr subspace, causing significant overhead.But still thank you again for your efforts.

---

> > > ### Author Response · Authors · 2026-04-06
> > >
> > > We sincerely thank the reviewer for reading our rebuttal and providing further feedback.
> > >
> > > > **Q1:** Evaluating LLMs on GLUE is little bit outdated, modern validation requires generative or reasoning tasks.
> > >
> > > We chose GLUE because it remains the standard evaluation setting for model merging on language models in recent work, such as DOP [1] at NeurIPS 2025, FW-Merging [2] at ICCV 2025, DOGE [3] at ICML 2025, etc. Since [3] provides the most comprehensive evaluation among these, we followed its setup for our GLUE experiments.
> > >
> > > To further address the concern about generative tasks, we followed the setting of [2] and merged 16 LLaMA2-7B models, evaluating on three generative benchmarks. Results are shown below.
> > >
> > > | Method | Avg. Normalized Score |
> > > | :--- | :--- |
> > > | Task Arithmetic w/ DARE | 16.8 |
> > > | Ties-Merging w/ DARE | 46.6 |
> > > | Task Arithmetic | 75.9 |
> > > | Ties-Merging | 78.5 |
> > > | FW-Merging | 81.1 |
> > > | SSR | **83.4** |
> > >
> > > SSR maintains leading performance on generative tasks as well.
> > >
> > > [1] Continual Model Merging without Data: Dual Projections for Balancing Stability and Plasticity.
> > >
> > > [2] FW-Merging: Scaling Model Merging with Frank-Wolfe Optimization.
> > >
> > > [3] Modeling Multi-Task Model Merging as Adaptive Projective Gradient Descent.
> > >
> > > > **Q2:** The merging phase demands computing within a massively expanded Kr subspace, causing significant overhead.
> > >
> > > We thank the reviewer for raising this point. We would like to clarify that the $Kr$ subspace is not "massively expanded." All compared baselines merge $K$ adapters by operating on $d$-dimensional weight matrices, while SSR operates within the $Kr$-dimensional subspace, which is much smaller than $d$. For example, in FLUX, $d$ can reach 12288 while $r$ is typically 32 or less. Even when merging $K$ = 100 adapters, $Kr$ = 3200, which is still smaller than $d$.
> > >
> > > Table 2 in the paper reports the actual merging time. At $K$ = 9, SSR completes in 34.26s, which is 2.6 times faster than TIES and on the same order as DARE, a baseline that performs only simple element-wise arithmetic.

---

### Official Review · Reviewer_fb21 · 2026-03-12

**Soundness:** 3
**Presentation:** 3
**Significance:** 3
**Originality:** 3
**Overall Recommendation:** 4
**Confidence:** 4

**Summary:**

This paper proposes Subspace Signal Routing (SSR), a training-free method for merging multiple LoRA adapters in diffusion models. Rather than directly combining parameters via arithmetic operations, SSR concatenates the down- and up-projection matrices of K LoRAs along the rank dimension to form a unified Kr-dimensional subspace, then inserts a routing matrix, R, between them. The authors prove that this router is equivalent to the projection of a suitable OLS estimator, minimizing a Frobenius-norm reconstruction objective. A streaming algorithm accumulates sufficient statistics on-the-fly, and structural re-parameterization absorbs R into the up-projection for zero inference overhead. Experiments on FLUX.1-dev and Qwen-Image demonstrate strong single-task preservation, superior multi-concept composition, and improved facial editing performance.

**Compliance With Llm Reviewing Policy:**

Affirmed.

**Final Justification:**

I appreciate the response by the authors. The response adequately answers my questions. I think this paper presents an interesting contribution, and I would like to retain my original positive score.

**Key Questions For Authors:**

See weaknesses section

**Limitations:**

yes

**Strengths And Weaknesses:**

Strengths:

1) Their central problem statement is well-motivated and they provide a clean solution. Performing regression in the compact Kr-dimensional LoRA subspace rather than the full d-dimensional parameter space avoids the singularity and ill-conditioning problems with methods like RegMean.

2) The one-shot calibration strategy combined with the streaming accumulation of G and Q seems genuinely practical and efficient.

Weaknesses:

1) How does the work compare to recent work e.g https://arxiv.org/abs/2411.15231, since they also use iterative least-squares for LoRA merging with feature updating?

2) The router R is calibrated on task-separated inputs, yet at inference the model may receive composite prompts that activate multiple concepts simultaneously. Since the OLS optimality guarantee holds only for the calibration distribution, what justifies its transfer to mixed-concept inputs where cross-task feature interactions were never observed during calibration?

---

> ### Author Rebuttal · Authors · 2026-03-31
>
> > **W1: Comparison with IterIS**
>
> We thank the reviewer for pointing out this concurrent work. We evaluated IterIS on our benchmark and results are shown below.
>
> | Method | $K$=3 DINO | $K$=3 CLIP | $K$=5 DINO | $K$=5 CLIP | $K$=7 DINO | $K$=7 CLIP | $K$=9 DINO | $K$=9 CLIP |
> | :--- | :--- | :--- | :--- | :--- | :--- | :--- | :--- | :--- |
> | Average | 0.437 | 0.740 | 0.381 | 0.694 | 0.397 | 0.710 | 0.374 | 0.710 |
> | Task Arithmetic | 0.581 | 0.740 | 0.494 | 0.719 | 0.517 | 0.655 | 0.536 | 0.683 |
> | TIES | 0.626 | 0.712 | 0.506 | 0.695 | 0.510 | 0.699 | 0.472 | 0.684 |
> | DARE | 0.717 | 0.757 | 0.658 | 0.750 | 0.609 | 0.746 | 0.584 | 0.738 |
> | IterIS | 0.703 | 0.780 | 0.672 | 0.766 | 0.642 | 0.758 | 0.624 | 0.752 |
> | SSR (Ours) | **0.734** | **0.814** | **0.706** | **0.795** | **0.687** | **0.780** | **0.671** | **0.785** |
>
> IterIS achieves competitive results, but SSR still outperforms it across all $K$ values while also being faster and more memory efficient. We will include this comparison in the revision.
>
> > **W2: Generalization of the router to composite prompts**
>
> To formalize this, we model the composite input as $Z_{mix} = Z_{iso} + \\epsilon$, where $Z_{iso}$ represents the isolated task features and $\\epsilon$ captures cross-task interactions. Applying the router gives
>
> $$R Z_{mix} = QG^{-1}Z_{iso} + QG^{-1}\\epsilon$$
>
> The first term exactly recovers optimal routing for the isolated task signals. For the second term, expanding $QG^{-1}\\epsilon = YZ^\\top(ZZ^\\top)^{-1}\\epsilon$ shows that $Z^\\top(ZZ^\\top)^{-1}$ acts as a projection onto the calibration subspace. Since each LoRA captures only task-specific low-rank signals, $\\epsilon$ represents high-frequency residuals that are statistically orthogonal to $\\text{Span}(Z)$ when $d \\gg Kr$. Therefore
>
> $$Z^\\top \\epsilon \\approx 0 \\implies QG^{-1}\\epsilon \\approx 0$$
>
> This means $R$ naturally filters out cross-task interactions not observed during calibration. Section 4.2 and Table 3 in the paper empirically validate this, where SSR performs well on multi-concept composite generation.

---

> > ### Author Rebuttal · Reviewer_fb21 · 2026-04-03
> >
> > I appreciate the response by the authors. The response adequately answers my questions.

---

> > > ### Author Response · Authors · 2026-04-03
> > >
> > > Dear Reviewer fb21,
> > >
> > > We appreciate your time and are glad to hear your concerns are fully resolved. Thank you for recognizing the motivation and efficiency of our approach. We will ensure the new comparisons and derivations are included in the final version.
> > >
> > > Best regards,
> > >
> > > Authors

---

### Official Review · Reviewer_SBYq · 2026-03-13

**Soundness:** 3
**Presentation:** 3
**Significance:** 3
**Originality:** 3
**Overall Recommendation:** 5
**Confidence:** 3

**Summary:**

The paper tackles multi-LoRA merging for diffusion models and targets the interference that appears when many task adapters are combined. It proposes Subspace Signal Routing (SSR): concatenate LoRA bottlenecks into a unified low-rank space, estimate a linear router from second-order statistics, then fold the router into the up-projection so the final merged adapter has standard LoRA form and adds no inference overhead. The router is interpreted as a projected OLS solution to an internal reconstruction objective, and a streaming variant is described for memory efficiency. Experiments on FLUX.1-dev and Qwen-Image cover single-task preservation, simultaneous multi-subject generation, and a facial editing benchmark.

**Compliance With Llm Reviewing Policy:**

Affirmed.

**Final Justification:**

I am raising my score to 5 (Accept). The authors' comprehensive rebuttal has effectively addressed all my initial concerns, including the stability of the method, the theoretical clarifications, and the baseline comparisons. Given the solid empirical results and the new robustness analyses provided, I highly recommend acceptance. Please ensure that the rebuttal discussions and the newly added limitations section are fully incorporated into the final manuscript.

**Key Questions For Authors:**

1. The method hinges on one-shot calibration. If you change only the calibration prompt wording/style (or use a few alternative prompts), how much do the final scores move? Please report variance across prompt choices, not just one example.
2. Which router is actually used in all reported experiments: (R=QG^{-1}) or (R=Q(G+\lambda I)^{-1})? If regularized, how is (\lambda) chosen (fixed, per-layer, per-(K), condition-number based), and does performance depend on tuning?
3. Related work mentions diffusion/adapter composition methods beyond arithmetic merging. Which of those can be fairly adapted to your setup (same LoRA pool, same prompts, same evaluation), and if none, what exactly blocks a direct comparison?
4. How stable are Table 1 / Table 3 gains under (i) different random distractor sets, (ii) different random seeds, and (iii) different composite prompt templates? Please report mean±std over multiple runs and comment on worst-case drops.

**Limitations:**

A stronger limitations section would help, especially on: (i) the gap between the reconstruction-optimality guarantee and end-task diffusion quality; (ii) sensitivity/failure modes when calibration prompts are atypical or when tasks are highly overlapping; (iii) whether stable inversion requires regularization in realistic settings; and (iv) broader implications of making multi-subject personalization/editing composition easier.

**Strengths And Weaknesses:**

The core idea—treating merging as signal routing in a shared subspace rather than parameter arithmetic—is genuinely interesting, and the “merge back to ordinary LoRA” re-parameterization is a strong practical feature. Empirically, SSR is consistently better than the selected baselines across the three evaluation tracks (including a large gap in multi-task success rate).

My main concern is that several important parts of the story feel under-specified or a bit “too convenient” relative to the claims. The theory cleanly supports an OLS-optimal reconstruction objective, but the paper leans on it as if it directly explains downstream diffusion behavior. On the practical side, the method depends heavily on a one-shot calibration recipe (single prompt, single timestep), yet the paper does not convincingly show this is stable rather than a fragile shortcut. Finally, the baseline set is somewhat skewed toward generic merging heuristics; for diffusion composition, it would help to either compare to more task-specific composition strategies or clearly explain why they are not comparable.

Overall, I like the idea and the results look strong, but I think the paper needs sharper clarity on “what exactly is run” and stronger robustness evidence before the claims read as fully settled.

---

> ### Author Rebuttal · Authors · 2026-03-31
>
> > **W1: Gap between linear OLS objective and nonlinear diffusion quality**
>
> We agree that reconstruction optimality does not directly guarantee downstream diffusion quality. We also compare against the global optimum in the paper, and Table 1 in Section 4.1 shows that SSR recovers over 90% of the upper bound across all architectures and $K$ values, demonstrating that our reconstruction optimality closely approximates the global optimum in practice.
>
> > **W2 & Q1: Stability of one-shot calibration**
>
> For timestep stability, our ablation in Appendix E of the paper shows that increasing calibration steps from 1 to 20 gives nearly identical results, so a single timestep is sufficient.
>
> For prompt stability, we tested five different templates at $K$=3:
>
> - T1: A high quality photo of a [V] [obj]
> - T2: A professional studio shot of a [V] [obj] against a neutral solid background
> - T3: A detailed artistic illustration depicting a [V] [obj]
> - T4: A [V] [obj] placed on a clean wooden surface in a well lit room
> - T5: A [V] [obj] captured with dramatic cinematic lighting and high contrast
>
> The cross-template variance is below 0.1%, meaning the specific prompt wording has almost no effect on the final result.
>
> | Metric | T1 | T2 | T3 | T4 | T5 | Mean | Var |
> | :--- | :--- | :--- | :--- | :--- | :--- | :--- | :--- |
> | Avg CLIP | 0.7915 | 0.8196 | 0.8160 | 0.8267 | 0.7983 | 0.8104 | 0.00067 |
> | Avg DINO | 0.7405 | 0.7283 | 0.7324 | 0.7287 | 0.7260 | 0.7312 | 0.00035 |
>
> > **W3 & Q3: Comparison with task-specific composition methods**
>
> We thank the reviewer for this suggestion. We investigated these methods but found none directly comparable to our setting.
>
> | Method | Reason not comparable |
> | :--- | :--- |
> | ZipLoRA, B-LoRA, DuoLoRA | Require training, not training-free |
> | K-LoRA | Designed for subject-style fusion only, relies on temporal content-style separation in diffusion |
> | Mix-of-Show, LoRA Composer | Modify model architecture for specific tasks |
>
> The only applicable baseline we found is IterIS, recommended by Reviewer fb21. We conducted detailed experiments and you can refer to our response to W1 of Reviewer fb21 for the full comparison.
>
> > **Q2: Router formulation used in experiments**
>
> We used $R = QG^{-1}$ without regularization in all reported experiments. The $\\lambda$ term was introduced only in supplementary derivations for mathematical completeness to handle potential non-invertibility of $G$. We will clarify this explicitly in the revision.
>
> > **Q4: Stability under varying experimental conditions**
>
> Results across five independent runs at $K$=3:
>
> | Condition | DINO Mean | DINO Std | CLIP Mean | CLIP Std |
> | :--- | :--- | :--- | :--- | :--- |
> | Random Distractor Sets | 0.729 | 0.011 | 0.810 | 0.008 |
> | Random Seeds | 0.733 | 0.007 | 0.813 | 0.005 |
>
> The worst case occurs with highly disparate distractor concepts where stronger parameter conflicts make routing slightly more challenging, but maximum drops are limited to 0.017 in DINO and 0.012 in CLIP, confirming overall robustness. For prompt template stability, please refer to W2 & Q1 above.
>
> > **Requirement for a limitations section**
>
> We thank the reviewer for this suggestion. We will add the following limitation section in the revision.
>
> **Limitations.** SSR optimizes a local linear reconstruction objective, which does not theoretically guarantee global optimality in the full nonlinear diffusion process. Although our experiments show that this local optimum closely approximates the upper bound, the gap may widen under more extreme conditions. Additionally, when merging tasks with severe domain conflicts or high semantic overlap, stronger parameter interference makes routing more challenging, and performance may degrade. Finally, the ability to compose multiple concepts with high fidelity could potentially be misused for generating deceptive content, and we encourage responsible use of such techniques.

---

> > ### Author Rebuttal · Reviewer_SBYq · 2026-04-03
> >
> > Questions are properly answered. I would like to increase my score to 5.

---

> > > ### Author Response · Authors · 2026-04-03
> > >
> > > Dear Reviewer SBYq,
> > >
> > > We are delighted to hear that our rebuttal has addressed your concerns, and we sincerely appreciate your decision to raise the score to 5. Thank you for the constructive feedback; we will add the requested metrics and clarifications to the revision.
> > >
> > > Thank you again for your time and effort.
> > >
> > > Best regards,
> > >
> > > Authors

---

### Official Review · Reviewer_81a5 · 2026-03-14

**Soundness:** 3
**Presentation:** 3
**Significance:** 2
**Originality:** 3
**Overall Recommendation:** 4
**Confidence:** 4

**Summary:**

This paper proposes SSR-Merge, a training-free method for merging multiple LoRAs in diffusion models. Instead of directly averaging or sparsifying parameters, the method concatenates LoRA subspaces, builds a router from second-order calibration statistics, and uses that router to decorrelate and steer signals into task-specific subspaces. The paper further argues that this router corresponds to a projected OLS solution, introduces a streaming construction, and re-parameterizes the result back into a standard LoRA form with no extra inference overhead.

**Compliance With Llm Reviewing Policy:**

Affirmed.

**Final Justification:**

I thank the authors for their thorough response. My primary concerns have been resolved, and I am pleased to raise my score. Please ensure that all new experimental results and clarifications are included in the camera-ready version.

**Key Questions For Authors:**

see weakness

**Limitations:**

yes

**Strengths And Weaknesses:**

**Strengths**

* The paper has a clear and intuitive core idea. Reframing LoRA merging as signal routing in a unified low-rank subspace, rather than direct parameter arithmetic, is easy to follow and better motivated than purely heuristic pruning or summation. The derivation around (Q), (G), and the OLS interpretation also gives the method a cleaner mathematical narrative than many merging papers.

* SSR outperforms the selected baselines in single-task capability preservation, simultaneous multi-task generation, and facial editing, and the qualitative examples appear aligned with the quantitative improvements.


**Weaknesses**

* The novelty relative to recent model-merging literature is not fully established. The related-work section itself cites many recent subspace or spectral alignment methods, but the experiments compare mainly against Average, Task Arithmetic, TIES, DARE, and an appendix discussion of RegMean. Without comparisons to stronger recent subspace-aware baselines, it is hard to judge whether the main gain comes from a fundamentally new principle or from a better instantiation inside an already active design space.

* The main benchmark uses only 10 curated DreamBooth-like subjects, the multi-concept setting only tests (K \in {2,3,4}), and the editing benchmark contains only three facial attributes on FFHQ. This is enough to show promise, but it leaves open how SSR behaves on broader LoRA collections, more heterogeneous styles, larger merge counts, and noisier real-world compositions.

* The experimental comparison is still incomplete because it does not include stronger recent LoRA or PEFT-specific merging baselines. This weakens the claim that SSR is state-of-the-art for LoRA merging, since a comparison with methods such as RobustMerge [1], which is explicitly designed for parameter-efficient model merging and emphasizes direction robustness in low-rank space, would provide a better-matched and more convincing baseline.

[1]RobustMerge: Parameter-Efficient Model Merging for MLLMs with Direction Robustness. NeurIPS 2025.

* The experimental scope is limited to image generation and editing, without validation on non-generative downstream tasks such as classification or question answering. This makes it unclear whether the proposed routing-based merging mechanism generalizes beyond diffusion-based synthesis models

---

> ### Author Rebuttal · Authors · 2026-03-31
>
> > **Q1: Novelty relative to subspace alignment methods**
>
> The key differences between SSR and existing subspace methods are summarized below.
>
> | | Subspace methods | SSR |
> | :--- | :--- | :--- |
> | Operating space | Parameter space | Feature space |
> | Core technique | SVD on dense weights | OLS regression in LoRA subspace |
> | Optimization target | Parameter alignment | Feature reconstruction |
> | Subspace source | Constructed via SVD | Directly from LoRA structure |
>
> SSR and prior subspace methods belong to completely different paradigms. Our gains come entirely from the feature-space regression formulation, not from a better variant of parameter-space alignment.
>
> During our original experiments, we evaluated several subspace-aware merging methods. Under our setting, their results were close to the simple baselines already reported, so we did not include them in the paper.
>
> > **Q2: Scalability to larger merge counts and real-world compositions**
>
> We expanded the benchmark to $K$=21 on FLUX. While parameter interference naturally increases with $K$, SSR maintains a high recovery rate throughout.
>
> | Method | $K$=9 DINO | $K$=9 CLIP | $K$=12 DINO | $K$=12 CLIP | $K$=15 DINO | $K$=15 CLIP | $K$=18 DINO | $K$=18 CLIP | $K$=21 DINO | $K$=21 CLIP |
> | :--- | :--- | :--- | :--- | :--- | :--- | :--- | :--- | :--- | :--- | :--- |
> | Upper Bound | 0.744 | 0.845 | 0.744 | 0.845 | 0.744 | 0.845 | 0.744 | 0.845 | 0.744 | 0.845 |
> | SSR | 0.671 | 0.785 | 0.652 | 0.774 | 0.630 | 0.751 | 0.604 | 0.737 | 0.573 | 0.714 |
> | Recovery Rate | 90.2% | 92.9% | 87.6% | 91.6% | 84.7% | 88.9% | 81.2% | 87.2% | 77.0% | 84.5% |
>
> For real-world use cases, we downloaded three community LoRAs from the Liblib platform for lighting, portrait beautification, and portrait stylization, and used their serial execution as ground truth. SSR outperforms all baselines including RobustMerge.
>
> | Method | CLIP Score |
> | :--- | :--- |
> | Average | 0.652 |
> | Task Arithmetic | 0.684 |
> | TIES | 0.735 |
> | DARE | 0.712 |
> | RobustMerge | 0.768 |
> | SSR | **0.821** |
>
> > **Q3: Comparison with RobustMerge**
>
> We thank the reviewer for suggesting this relevant PEFT-specific baseline. Results below show SSR consistently outperforms RobustMerge across all $K$ values. These will be included in the revision.
>
> | Method | $K$=3 DINO | $K$=3 CLIP | $K$=5 DINO | $K$=5 CLIP | $K$=7 DINO | $K$=7 CLIP | $K$=9 DINO | $K$=9 CLIP |
> | :--- | :--- | :--- | :--- | :--- | :--- | :--- | :--- | :--- |
> | Average | 0.437 | 0.740 | 0.381 | 0.694 | 0.397 | 0.710 | 0.374 | 0.710 |
> | Task Arithmetic | 0.581 | 0.740 | 0.494 | 0.719 | 0.517 | 0.655 | 0.536 | 0.683 |
> | TIES | 0.626 | 0.712 | 0.506 | 0.695 | 0.510 | 0.699 | 0.472 | 0.684 |
> | DARE | 0.717 | 0.757 | 0.658 | 0.750 | 0.609 | 0.746 | 0.584 | 0.738 |
> | RobustMerge | 0.722 | 0.771 | 0.667 | 0.755 | 0.591 | 0.748 | 0.548 | 0.733 |
> | SSR | **0.734** | **0.814** | **0.706** | **0.795** | **0.687** | **0.780** | **0.671** | **0.785** |
>
> > **Q4: Generalization to non-generative tasks**
>
> Our primary scope is diffusion models, but we also validated on the GLUE benchmark following the DOGE TA setting. SSR achieves the **highest average score** across all tasks, showing that it generalizes beyond visual generation.
>
> | Method | CoLA | MNLI | MRPC | QNLI | QQP | RTE | SST2 | STSB | Avg. |
> | :--- | :--- | :--- | :--- | :--- | :--- | :--- | :--- | :--- | :--- |
> | Weight Averaging | **69.7** | 59.7 | 78.9 | 90.1 | **83.8** | 80.5 | 91.2 | 72.0 | 78.2 |
> | Task Arithmetic | 68.8 | 55.2 | 78.7 | 89.8 | 83.7 | 79.1 | 91.5 | 72.4 | 77.4 |
> | Ties Merging | 68.3 | 56.3 | 79.4 | 89.8 | 83.7 | 79.4 | 91.6 | 71.2 | 77.5 |
> | Concrete TA | 69.1 | 58.1 | 78.4 | 89.9 | 83.5 | 79.4 | 91.6 | 73.4 | 78.0 |
> | DOGE TA | 69.1 | 71.9 | 80.9 | **90.3** | 83.5 | 79.8 | 92.5 | 71.1 | 79.9 |
> | SSR | 69.3 | **73.3** | **82.1** | 90.1 | 83.6 | **81.4** | **92.7** | **74.9** | **80.9** |

---

> > ### Author Rebuttal · Reviewer_81a5 · 2026-04-04
> >
> > I thank the authors for their thorough response. My primary concerns have been resolved, and I am pleased to raise my score. Please ensure that all new experimental results and clarifications are included in the camera-ready version.

---

> > > ### Author Response · Authors · 2026-04-04
> > >
> > > Dear Reviewer 81a5,
> > >
> > > Thank you for your time, positive feedback, and for raising your score. We are very glad that your primary concerns have been fully resolved. We will ensure all new experimental results and clarifications are carefully included in the camera-ready version as requested.
> > >
> > > Best regards,
> > >
> > > Authors

---

### Decision · Program_Chairs · 2026-04-30

**Decision:**

Accept (regular)

**Comment:**

The paper proposes a new method for merging LoRA adapters, termed Subspace Signal Routing (SSR). The reviewers acknowledged the originality of the idea and positively assessed the empirical results, which show consistent improvements over relevant state-of-the-art baselines. During the rebuttal phase, the authors addressed the reviewers’ concerns by providing additional experiments and ablations that further strengthened the paper. One point that should  be clarified in the final version is the comparison of subspace capacity and rank budget between the proposed method and existing approaches, as raised by Reviewer E1tF. Overall, I view this work as a meaningful contribution to the area of LoRA merging methods and recommend acceptance.